# Offline RL by Reward-Weighted Fine-Tuning for Conversation Optimization

**Subhojyoti Mukherjee, Viet Dac Lai, Raghavendra Addanki, Ryan Rossi**
Adobe Research
subhomuk@adobe.com

**Seunghyun Yoon, Trung Bui, Anup Rao, Jayakumar Subramanian, Branislav Kveton**
Adobe Research

## Abstract

Offline reinforcement learning (RL) is a variant of RL where the policy is learned from a previously collected dataset of trajectories and rewards. In our work, we propose a practical approach to offline RL with large language models (LLMs). We recast the problem as reward-weighted fine-tuning, which can be solved using similar techniques to supervised fine-tuning (SFT). To showcase the value of our approach, we apply it to learning short-horizon question-answering policies of a fixed length, where the agent reasons about potential answers or asks clarifying questions. Our work stands in a stark contrast to state-of-the-art methods in this domain, based on SFT and direct preference optimization, which have additional hyper-parameters and do not directly optimize for rewards. We compare to them empirically, and report major gains in both optimized rewards and language quality.

## 1 Introduction

*Reinforcement learning (RL)* [62] is a machine learning framework for learning to act sequentially in an unknown environment with the goal of maximizing a long-term reward. Because of its generality and broad applicability, RL has been studied extensively and many RL algorithms have been proposed, including temporal-difference learning [61], Q-learning [70], policy gradients [72], and actor-critic methods [63]. All of these algorithms learn from online interactions with the environment, which is often not possible due to engineering and safety constraints. This motivates the need for *offline reinforcement learning* [34, 81]. The key idea in offline RL is to collect a dataset of interactions with the environment and learn a policy from it, akin to learning a classifier in supervised learning [35]. Offline RL is especially suitable for problems where offline interactions are abundant or can be easily simulated. For instance, *question answering (QA)* is an area at the intersection of *natural language processing (NLP)* and information retrieval concerned with building systems for answering natural language questions. Since many QA datasets exist [71, 43, 14, 22] and QA can be simulated using pre-trained *large language models (LLMs)* [51, 7, 6], several recent works on QA focused on learning better QA policies using offline RL.

The main contribution of our work are two novel algorithms for offline RL with LLMs: `Refit` and `Swift`. The key idea in `Refit` is to optimize a lower bound on the online RL objective, which is the sum of the log-probabilities of logged actions weighted by trajectory rewards. This new objective has two main benefits. First, it does not involve ratios of token-level propensity scores, unlike in PPO [57] and GRPO [58]. This leads to a stable and practical algorithm. Second, the optimization of the objective can be viewed as *weighted fine-tuning* and solved by a minor modification of *supervised fine-tuning (SFT)*, a standard post-training technique for LLMs. Motivated by GRPO [58], we also propose `Swift`, which is a variant of `Refit` where we standardize trajectory rewards using multiple

39th Conference on Neural Information Processing Systems (NeurIPS 2025).

logged trajectories. The standardized rewards lower the variance in policy optimization and may improve learned policies, as we show empirically in Section 4. We try to justify this theoretically in Appendix A.3.

To show the value of `Refit` and `Swift`, we apply them to learning multi-turn QA policies, where the agent reasons about potential answers or asks clarifying questions. The closest related works are Andukuri et al. [2] and Chen et al. [11], which use RL to learn clarifying questions from simulated agent-teacher conversations. Andukuri et al. [2] choose the most rewarding trajectories and fine-tune on them. Chen et al. [11] generate alternative responses for each step of the conversation and then optimize for better responses using DPO [52]. The main limitation of these approaches is that they do not fully utilize the reward signal; they only use it to turn the original problem into an SFT or DPO problem. We directly optimize for rewards using RL. We observe major gains over SFT and DPO in experiments (Section 4) because an indirect optimization of rewards results in information loss.

We make the following contributions:

1. We formulate a generic RL problem that encompasses conversation optimization and does not make strong assumptions on rewards. Our setting captures fixed-horizon conversations and adaptive ones, when the conversation can stop at any time, for instance because enough information to answer the question has been gathered.

2. We derive an offline RL objective, which is a lower bound on the online RL objective. As a result, the online objective is optimized by maximizing the offline one. The offline objective is equivalent to weighted fine-tuning and thus can be optimized in LLMs using standard SFT training primitives. The weights are over sequences of tokens, unlike individual tokens in prior works [53, 18, 76, 80]. We also avoid propensity score ratios [57, 58].

3. We derive an offline RL objective with standardized rewards. The standardized rewards lower the variance in policy optimization and may improve learned policies. We show this empirically in Section 4.

4. We comprehensively evaluate our approach on multi-turn QA problems over datasets spanning open book exams, textual information for science topics, conversational text-to-SQL, and mathematical dialogue and problem solving. Although we optimize a single reward, we observe improvements in all other metrics, such as reasoning ability, pedagogical value, and confidence. We consider five baselines: two variants of SFT, two variants of DPO, and the original policy. We observe major gains over SFT and DPO because we directly optimize for rewards using RL.

5. For each QA benchmark, we generate a dataset of 500 multi-turn conversations. We give instructions for reproducing the datasets, including all prompts and example conversations, in Appendices E and F.

The paper is organized as follows. We present our setting in Section 2. In Section 3, we formulate our offline RL objectives and show how to optimize them using weighted fine-tuning. We report our results in Section 4 and discuss related work in Section 5. We conclude in Section 6.

## 2 Setting

We start with introducing our notation. We denote the marginal and conditional probabilities under the probability measure $p$ by $p(X = x)$ and $p(X = x \mid Y = y)$, respectively; and write $p(x)$ and $p(x \mid y)$ when the random variables are clear from context. The indicator function is $\mathbb{1}\{\cdot\}$. For a positive integer $n$, we define $[n] = \{1, \dots, n\}$. The $i$-th entry of vector $v$ is $v_i$. If the vector is already indexed, such as $v_j$, we write $v_{j,i}$.

We view the problem of learning multi-turn conversation policies as a generic reinforcement learning problem [62] where an *agent* interacts with an *environment*. The agent takes actions conditioned on the conversation history and the environment responds. When the conversation ends, it is given a reward. The reward measures the quality of the conversation and the agent maximizes it.

We formalize the problem as follows. The agent first observes *context* $x \in \mathcal{S}$, where $\mathcal{S}$ is the space of all strings, each represented as a sequence of tokens. The context defines the task. The conversation between the agent and environment consists of steps indexed by $t \in \mathbb{N}$, where $\mathbb{N}$ is a set of positive

integers. In step $t$, the agent takes an *action* $a_t \in \mathcal{S}$ and the environment responds with an *observation* $y_t \in \mathcal{S}$. The conversation is a *trajectory* $\tau_n = (a_1, y_1, \ldots, a_n, y_n)$ of $n$ actions and observations. The number of steps $n$ can be fixed or random. When $n$ is random, the conversation end can be any function of the conversation history. The *reward* is a non-negative function of $x$ and $\tau_n$, denoted by $r(x, \tau_n) \geq 0$, that measures conversation quality. We do not assume that it factors over individual steps, as is common in RL. This is to maintain generality and because our algorithms (Section 3) do not need it.

The agent follows a policy conditioned on the conversation history. The probability that action $a$ is taken in context $x$ and history $\tau_{t-1}$ is denoted by $\pi(a \mid x, \tau_{t-1}; \theta)$ and parameterized by $\theta \in \Theta$. We call $\theta$ a *policy* and $\Theta$ the space of policy parameters. The probability of observing $y_t$ conditioned on conversation history $\tau_{t-1}$ and action $a_t$ is denoted by $p(y_t \mid x, \tau_{t-1}, a_t)$. We slightly abuse our notation and denote the probability of trajectory $\tau_n$ in context $x$ under policy $\theta$ by

$$\pi(\tau_n \mid x; \theta) = \prod_{t=1}^{n} p(y_t \mid x, \tau_{t-1}, a_t)\, \pi(a_t \mid x, \tau_{t-1}; \theta)\,. \tag{1}$$

The factorization follows from the chain rule of probability. The expected value of policy $\theta$, where $q$ is a distribution over contexts $x$, is

$$V(\theta) = \mathbb{E}_{x \sim q,\, \tau_n \sim \pi(\cdot \mid x; \theta)} \left[ r(x, \tau_n) \right]\,. \tag{2}$$

Our goal is to learn a policy that maximizes it, $\theta_* = \arg\max_{\theta \in \Theta} V(\theta)$.

Our framework is sufficiently general to model multiple use cases. For instance, suppose that we want to maximize the pedagogical value of a conversation over $n$ steps [55]. Then $r(x, \tau_n)$ would be the aggregated pedagogical value of $\tau_n$ over $n$ steps. As another example, suppose that we want to learn to clarify an ambiguous question $x$ by asking $n$ questions [2, 11]. Then $r(x, \tau_n)$ would be the quality of the generated answer conditioned on $x$ and $\tau_n$. Finally, suppose that the number of clarifying questions is adaptively chosen by the agent, after enough information has been gathered [33]. Then $r(x, \tau_n)$ would be the quality of the generated answer discounted by $\gamma^n$ for $\gamma \in (0, 1)$. The discounting prevents the agent from asking clarifying questions indefinitely since the reward diminishes with the number of steps $n$. In this case, the number of clarifying questions $n$ is random and decided by the agent.

## 3 Algorithms

Our objective is to maximize the expected policy value $V(\theta)$ in (2). This can be done in a myriad of ways [62]. The most natural approach for complex policies, like those represented by LLMs, are policy gradients [72]. The key idea in *policy gradients* is to update the policy $\theta$ iteratively by gradient ascent. The gradient of $V(\theta)$ at $\theta$ is

$$\nabla V(\theta) = \mathbb{E}_{x \sim q,\, \tau_n \sim \pi(\cdot \mid x; \theta)} \left[ r(x, \tau_n) \nabla \log \pi(\tau_n \mid x; \theta) \right]$$

and can be derived by a direct application of the score identity [1]. The computation of this gradient is challenging in real-world problems for two reasons. First, since the trajectories $\tau_n$ are sampled under the optimized policy $\theta$, they need to be resampled when $\theta$ changes, in each step of gradient ascent. Second, a reward model $r(x, \tau_n)$ is needed to evaluate any potentially sampled trajectory.

To address these challenges, we resort to *offline reinforcement learning* [34, 81, 35]. The key idea in offline RL is to collect a dataset of trajectories and their rewards once, and then learn a policy from it, akin to learning a classifier in supervised learning. We denote the *data logging policy* by $\pi_0$ and the probability of generating a trajectory $\tau_n$ in context $x$ using policy $\pi_0$ by $\pi_0(\tau_n \mid x)$. A classic result in control [16] and statistics [25] is that propensity scores,

$$V(\theta) = \mathbb{E}_{x \sim q,\, \tau_n \sim \pi(\cdot \mid x; \theta)} \left[ r(x, \tau_n) \right] = \mathbb{E}_{x \sim q,\, \tau_n \sim \pi_0(\cdot \mid x)} \left[ \frac{\pi(\tau_n \mid x; \theta)}{\pi_0(\tau_n \mid x)} r(x, \tau_n) \right]\,, \tag{3}$$

can correct for selection bias in the logged dataset. Simply put, the optimization of (2) is equivalent to maximizing propensity-weighted rewards on a dataset of trajectories collected by another policy $\pi_0$. One challenge with optimizing (3) is that the variance of the empirical estimate of (3) can be high when $\pi_0(\tau_n \mid x)$ is small. This can be addressed by clipping [29] at a token level, which is the key idea in PPO [57] and GRPO [58]. We discuss differences from these methods in more detail at the end of Section 3.1. Another challenge is that the data logging policy $\pi_0$ is often unknown. Now we present our approach of reward-weighted fine-tuning for offline RL.

### 3.1 Reward-Weighted Fine-Tuning

The key idea in our work is to maximize a lower bound on (2). While this bound is tight only when $\pi(\cdot \mid \cdot; \theta) \equiv \pi_0$, it leads to a practical offline RL algorithm that can be implemented using weighted fine-tuning *without introducing propensity score ratios*. We build on the lower bound in Ma et al. [41] and Liang and Vlassis [37], and extend it to offline RL.

**Lemma 1.** *For any policies $\pi$ and $\pi_0$, and any non-negative reward function,*

$$\mathbb{E}_{x \sim q, \, \tau_n \sim \pi(\cdot \mid x; \theta)} \left[ r(x, \tau_n) \right] \geq \mathbb{E}_{x \sim q, \, \tau_n \sim \pi_0(\cdot \mid x)} \left[ r(x, \tau_n) \log \pi(\tau_n \mid x; \theta) \right] + C_1 \,,$$

*where $C_1 = \mathbb{E}_{x \sim q, \, \tau_n \sim \pi_0(\cdot \mid x)} \left[ r(x, \tau_n)(1 - \log \pi_0(\tau_n \mid x)) \right] \geq 0$ is a constant independent of $\theta$.*

*Proof.* Using basic algebra,

$$\begin{aligned}
\mathbb{E}_{x \sim q, \, \tau_n \sim \pi(\cdot \mid x; \theta)} \left[ r(x, \tau_n) \right] &= \mathbb{E}_{x \sim q, \, \tau_n \sim \pi_0(\cdot \mid x)} \left[ r(x, \tau_n) \frac{\pi(\tau_n \mid x; \theta)}{\pi_0(\tau_n \mid x)} \right] \\
&\geq \mathbb{E}_{x \sim q, \, \tau_n \sim \pi_0(\cdot \mid x)} \left[ r(x, \tau_n) \left( 1 + \log \frac{\pi(\tau_n \mid x; \theta)}{\pi_0(\tau_n \mid x)} \right) \right] \\
&= \mathbb{E}_{x \sim q, \, \tau_n \sim \pi_0(\cdot \mid x)} \left[ r(x, \tau_n) \log \pi(\tau_n \mid x; \theta) \right] + C_1 \,.
\end{aligned}$$

The inequality follows from $u \geq 1 + \log u$ and non-negative rewards. $\qquad\square$

The bound is loose in practice because we apply $u \geq 1 + \log u$ for a potentially large $u$. The result of Lemma 1 is that

$$J(\theta) = \mathbb{E}_{x \sim q, \, \tau_n \sim \pi_0(\cdot \mid x)} \left[ r(x, \tau_n) \log \pi(\tau_n \mid x; \theta) \right] \tag{4}$$

is a lower bound on (2). Since the lower bound is tight when $\pi(\cdot \mid \cdot; \theta) \equiv \pi_0$, a policy that improves (4) also improves (2). Next we show that (4) is equivalent to reward-weighted fine-tuning. To see this, we plug the definition of the trajectory probability (1) into (4) and get

$$J(\theta) = \mathbb{E}_{x \sim q, \, \tau_n \sim \pi_0(\cdot \mid x)} \left[ r(x, \tau_n) \sum_{t=1}^{n} \log \pi(a_t \mid x, \tau_{t-1}; \theta) \right] + C \,, \tag{5}$$

where $C = \mathbb{E}_{x \sim q, \, \tau_n \sim \pi_0(\cdot \mid x)} \left[ r(x, \tau_n) \sum_{t=1}^{n} \log p(y_t \mid x, \tau_{t-1}, a_t) \right]$ represents the log-probabilities of observations weighted by trajectory rewards. Because the observation probabilities do not depend on $\theta$ (Section 2) and neither does $\tau_n \sim \pi_0(\cdot \mid x)$, $C$ is a constant independent of $\theta$. As a result, the maximization of (5) is equivalent to maximizing $n$ log-probabilities of actions $a_t \mid x, \tau_{t-1}$ weighted by trajectory reward $r(x, \tau_n)$. Therefore, we maximize the likelihood of trajectories proportionally to their rewards, by equally attributing the reward to each action in the trajectory. Our objective can also be viewed as weighted fine-tuning with $n$ terms. The terms are correlated because they belong to the same trajectory and are weighted by the same reward.

**PPO, GRPO, and Q-SFT.** We compare (5) to other RL objectives in LLMs next. Let $a_{t,i}$ be the $i$-th token in action $a_t$ and $a_{t,<i}$ be the first $i-1$ tokens in action $a_t$. Then the objective of PPO [57] in our problem can be written as

$$\mathbb{E}_{x \sim q, \, \tau_n \sim \pi_0(\cdot \mid x)} \left[ \sum_{t=1}^{n} \sum_{i} \min\{ P_{t,i} A_{t,i}, \, \mathrm{clip}(P_{t,i}, 1 - \epsilon, 1 + \epsilon) A_{t,i} \} \right] \,, \tag{6}$$

where $P_{t,i} = \pi(a_{t,i} \mid x, \tau_{t-1}, a_{t,<i}; \theta) / \pi_0(a_{t,i} \mid x, \tau_{t-1}, a_{t,<i})$ is the ratio of token-level propensity scores for the $i$-th token in action $a_t$, $A_{t,i}$ is the corresponding advantage, and $\mathrm{clip}$ clips propensity scores to $[1 - \epsilon, 1 + \epsilon]$ for some $\epsilon \in [0, 1]$. Our objective is different in two aspects. First, (5) does not involve token-level propensity score ratios, which can be large and cause numerical instability. In PPO, this is typically mitigated by tuning $\epsilon$. Second, the computation of the advantage $A_{t,i}$ requires a token-level reward model [56]. GRPO [58] can be viewed as PPO where $A_{t,i}$ in (6) is estimated using standardized rewards obtained by simulation. So the main difference of (5) from GRPO is that it does not involve token-level propensity score ratios. Finally, Q-SFT of Hong et al. [24] optimizes $\sum_{t=1}^{n} \sum_{i} Q_{t,i} \log \pi(a_{t,i} \mid x, \tau_{t-1}, a_{t,<i}; \theta)$, where $Q_{t,i}$ is the Q-function for the $i$-th token in action $a_t$ that depends on its reward, the ratio of propensity scores for the next token, and maximization

over it. To summarize, our objective does not involve token-level propensity score ratios, which can be large and cause numerical instability.

`STaR-GATE` **and** `StepDPO`**.** Now we compare (5) to related works in conversation optimization using RL. These algorithms are the state of the art in our domain and we compare to them empirically in Section 4. Andukuri et al. [2] apply SFT to most rewarding trajectories, which can be viewed as replacing $r(x, \tau_n)$ in (5) with an indicator that the trajectory has a high reward. Chen et al. [11] learn to take the best action in each step by minimizing the DPO loss, which can be viewed as replacing each term in (5) with the negative DPO loss. We observe major empirical gains over both of these works because they do not fully utilize the reward signal; they only use it to turn the original problem into a corresponding SFT or DPO problem.

## 3.2 Algorithm `Refit`

Our algorithm is an iterative optimization of (5). We call it reward-weighted fine-tuning (`Refit`) and give its pseudo-code in Algorithm 1. The input to `Refit` is a dataset $\mathcal{D} = \{(x, \tau_n, r)\}$ collected by a data logging policy $\pi_0$. The dataset is generated as follows. First, we sample context $x \sim q$. Second, we sample trajectory $\tau_n \sim \pi_0(\cdot \mid x)$ and get its reward $r(x, \tau_n)$. Finally, we add $(x, \tau_n, r(x, \tau_n))$ to the dataset and repeat this process until $\mathcal{D}$ is generated.

The policy $\theta$ is optimized by gradient ascent. The gradient of $J(\theta)$ at $\theta$ is

$$\nabla J(\theta) = \mathbb{E}_{x \sim q, \, \tau_n \sim \pi_0(\cdot|x)} \left[ r(x, \tau_n) \sum_{t=1}^{n} \nabla \log \pi(a_t \mid x, \tau_{t-1}; \theta) \right] . \tag{7}$$

The optimization is iterative. In iteration $i$, we approximate $\nabla J(\theta)$ by the gradient $g_i$ on a single trajectory $(x, \tau_n, r) \in \mathcal{D}$. Since the trajectories are generated i.i.d., $g_i$ is an unbiased estimate of (7). After $g_i$ is computed, we update the policy as $\theta + \alpha_i g_i$, where $\alpha_i > 0$ is a learning rate. The optimization ends after a single pass over the dataset but more passes are possible. Since $g_i$ is algebraically equivalent to the gradient on $n$ SFT data points weighted by the same reward, we implement `Refit` by modifying SFT in TRL [66].

---

**Algorithm 1** `Refit` / `Swift`

---

1: **Input:** Learning rate schedule $(\alpha_i)_{i \in \mathbb{N}}$
2: Generate a logged dataset $\mathcal{D} = \{(x, \tau_n, r)\}$, where $r \in \mathbb{R}$ is a reward of $\tau_n$ (`Refit`) or a standardized reward of $\tau_n$ (`Swift`)
3: Initialize $\theta$ and $i \leftarrow 1$
4: **for all** $(x, \tau_n, r) \in \mathcal{D}$ **do**
5: $\quad g_i \leftarrow r \sum_{t=1}^{n} \nabla \log \pi(a_t \mid x, \tau_{t-1}; \theta)$
6: $\quad \theta \leftarrow \theta + \alpha_i g_i$ and $i \leftarrow i + 1$
7: **Output:** Learned policy $\theta$

---

Finally, note that in expectation, an update by gradient $r \sum_{t=1}^{n} \nabla \log \pi(a_t \mid x, \tau_{t-1}; \theta)$ is equivalent to fine-tuning on $a_t \mid x, \tau_{t-1}$ for $\lfloor r \rfloor$ times with probability $\lceil r \rceil - r$ and for $\lceil r \rceil$ times otherwise. As a result, `Refit` can be trivially implemented in closed models through an SFT dataset, where each $a_t \mid x, \tau_{t-1}$ appears either $\lfloor r \rfloor$ or $\lceil r \rceil$ times, after randomized rounding.

## 3.3 Standardized Reward-Weighted Fine-Tuning

One challenge with (7) is that the empirical variance of the estimator can be high. As an example, suppose that the rewards are in $[9, 10]$. Then the gradient would be scaled by 10 instead of 1, when we subtract 9 from all rewards. This motivated many prior works on variance reduction in policy gradients [63, 5, 46]. This also motivates our work on optimizing standardized rewards. We start by showing that the optimization of standardized rewards is equivalent to optimizing (2) under certain assumptions.

**Lemma 2.** *Let $\mu(x) \geq 0$ and $\sigma(x) > 0$ be any non-negative functions of context $x$. Let $\tilde{r}(x, \tau_n) = (r(x, \tau_n) - \mu(x))/\sigma(x)$ be the* standardized reward*. Suppose that there exists $\theta_*$ that maximizes all $\mathbb{E}_{\tau_n \sim \pi(\cdot|x;\theta)} [r(x, \tau_n) \mid x]$ jointly. Then it also maximizes*

$$\mathbb{E}_{x \sim q, \, \tau_n \sim \pi(\cdot|x;\theta)} [\tilde{r}(x, \tau_n)] . \tag{8}$$

The proof is in Appendix A.1. The key assumption in Lemma 2, that there exists $\theta_*$ that maximizes all $\mathbb{E}_{\tau_n \sim \pi(\cdot|x;\theta)} [r(x, \tau_n) \mid x]$ jointly, is expected to be satisfied or near-satisfied when the policy class is rich, such as when represented by an LLM. This is because the policy is conditioned on $x$.

In the rest of this section, we derive an offline variant of (8) with similar desirable properties to (4) in Section 3.1. The challenge with applying the same reasoning is that the standardized rewards $\tilde{r}(x, \tau_n)$ can be negative. The error of our approximation is characterized below.

**Lemma 3.** *For any policies $\pi$ and $\pi_0$, and any rewards in $[-b, b]$,*

$$\left| \mathbb{E}_{x \sim q, \, \tau_n \sim \pi(\cdot|x;\theta)} \left[ \tilde{r}(x, \tau_n) \right] - \mathbb{E}_{x \sim q, \, \tau_n \sim \pi_0(\cdot|x)} \left[ \tilde{r}(x, \tau_n) \log \pi(\tau_n \mid x; \theta) \right] \right| \leq |C_1| + C_2 \,,$$

*where $C_1$ is a constant independent of $\theta$ defined in Lemma 1 and*

$$C_2 = b \max_{\theta \in \Theta, \, x, \, \tau_n} \left( \frac{\pi(\tau_n \mid x; \theta)}{\pi_0(\tau_n \mid x)} - \left( 1 + \log \frac{\pi(\tau_n \mid x; \theta)}{\pi_0(\tau_n \mid x)} \right) \right) \,.$$

The proof is in Appendix A.2. Lemma 3 says that the difference between the online objective in (8) and its offline counterpart

$$J(\theta) = \mathbb{E}_{x \sim q, \, \tau_n \sim \pi_0(\cdot|x)} \left[ \tilde{r}(x, \tau_n) \log \pi(\tau_n \mid x; \theta) \right] \tag{9}$$

is $O(|C_1| + C_2)$. While $C_2$ can be large, as it depends on the ratios of propensity scores, it is on the same order as the gap in Lemma 1. This is because the key step in the proof of Lemma 1 is that we apply $u \geq 1 + \log u$ for $u = \pi(\tau_n \mid x; \theta)/\pi_0(\tau_n \mid x)$. The main difference from Lemma 1 is that we do not get a proper lower bound. Using the same reasoning as in Section 3.1, the maximization of (9) is equivalent to fine-tuning on $n$ actions $a_t \mid x, \tau_{t-1}$ weighted by the standardized trajectory reward $\tilde{r}(x, \tau_n)$. The terms are correlated because they belong to the same trajectory and are weighted by the same reward.

We implement the optimization of (9) using Algorithm 1. The only difference is that the rewards are standardized and thus we call this method standardized reward-weighted fine-tuning (`Swift`). The logged dataset $\mathcal{D} = \{(x, \tau_n, \tilde{r})\}$ is generated as follows. First, we sample $x$. Second, we sample $m$ trajectories $\tau_{n,i} \sim \pi_0(\cdot \mid x)$ for $i \in [m]$ and compute their rewards $r(x, \tau_{n,i})$. Third, we estimate the mean reward $\mu(x)$ and the standard deviation of rewards $\sigma(x)$ as

$$\hat{\mu}(x) = \frac{1}{m} \sum_{i=1}^{m} r(x, \tau_{n,i}) \,, \quad \hat{\sigma}(x) = \sqrt{\frac{1}{m-1} \sum_{i=1}^{m} (r(x, \tau_{n,i}) - \hat{\mu}(x))^2} \,,$$

respectively. Finally, we standardize all rewards as $\tilde{r}(x, \tau_{n,i}) = (r(x, \tau_{n,i}) - \hat{\mu}(x))/\hat{\sigma}(x)$ and add all $(x, \tau_{n,i}, \tilde{r}(x, \tau_{n,i}))$ to the dataset. This process is repeated until $\mathcal{D}$ is generated. Note that the cost of the standardization, computing $\hat{\mu}(x)$ and $\hat{\sigma}(x)$, is $O(mn)$. So it is of the same order as sampling $m$ trajectories of length $n$ and thus negligible.

## 4 Experiments

We evaluate our methods on 6 datasets. OpenBookQA [43], ARC [14], SciQA [71], and MMLU [22] are standard QA benchmarks. We convert a text-to-SQL conversation dataset CoSQL [73] and math tutoring dataset MathDial [42] into QA-style conversational datasets. Our datasets cover a variety of domains and are described in more detail in Appendix D.

We generate 500 tasks for each dataset and report the average performance over the tasks per dataset. Each task is a conversation of length $n = 3$ between an agent represented by an *assistant* and the environment represented by a *teacher*. We experiment with two kinds of problems. In *reasoning experiments*, the teacher asks the assistant to solve the problem in step 1, encourages it to think deeper in step 2, and asks for a final answer in step 3. The prompts and conversation examples are reported in Appendix E. In *clarifying-questions experiments*, the assistant is also encouraged to ask questions and the teacher answers them. The prompts and conversation examples are reported in Appendix F. We experiment with both *thinking and standard modes*. The difference in the *thinking mode* is that the assistant reasons within <thinking> tags before responding. The assistant is implemented using Llama-3.1-8B-Instruct. In reasoning experiments, the teacher is scripted. In clarifying-questions experiments, the teacher is implemented using a combination of scripting and Llama-3.1-8B-Instruct. The model and training parameters are reported in Appendix G. We solve each task 3 times with different temperatures. The three runs are used for reward standardization in `Swift` and to implement our baselines [2, 11].

We report multiple metrics. Our *most fundamental* measure of performance is *Accuracy*, which is the proportion of questions whose answers match the correct (gold standard) answer. We report the percentage of times that the model outputs <thinking> tags as *Thinking*. This shows how well the model follows reasoning instructions. We also report six conversation *reward metrics* computed by a GPT-4o judge (Appendix E): **1.** *Overall*: A summary of the following five scores. **2.** *Accuracy*: Did the assistant select the correct answer? **3.** *Reasoning Ability*: Was the reasoning logical, clear, and precise? **4.** *Comprehensiveness*: Were alternative options addressed? **5.** *Pedagogical Value*: Would this explanation help someone to learn? **6.** *Confidence Calibration*: Was the assistant's confidence in giving the final answer appropriate? These metrics are reported with a prefix "R" in our tables. The reward in all RL algorithms is the overall reward rescaled to $[0, 1]$.

We consider five baselines. The first baseline is the original policy, and we call it `Base`. We expect to outperform `Base` due to learning. All other baselines are offline RL algorithms. To have a fair comparison, we use the same dataset of logged trajectories in all of them. The only difference is in how the dataset is used. `STaR-GATE` [2] learns policies by supervised fine-tuning on most rewarding trajectories. This is akin to reward signal thresholding, into the trajectories used for learning and not. We improve this baseline by distillation, as done in Andukuri et al. [2], and call it `STaR-GATE-D`. The fourth baseline is motivated by Chen et al. [11]. The key idea in Chen et al. [11] is to generate a new trajectory in each step of the original trajectories, and then determine winning and losing actions in that step based on the corresponding trajectory reward. After this, DPO is used to learn the winning actions. We call this baseline `StepDPO`. The main limitations of `STaR-GATE` and `StepDPO` are that they do not fully utilize the reward signal; they only use it to turn the original problem into an SFT or DPO problem. We directly optimize for rewards using RL. The last baseline is `DPO`, where the final winning and losing responses are used to directly answer the original question. This baseline shows what is possible without a conversation. We implement `Refit` and `Swift` as described in Section 3. We expect `Swift` to outperform `Refit` because reward-based learning tends to be sensitive to the scale of rewards [63, 5, 46].

**Reasoning experiments.** We report our results on all six datasets in Tables 1-12, in both thinking and standard modes. The best result is highlighted in **bold** and the second best result is underlined. The confidence intervals are standard errors of the estimates. The training times of all RL methods are comparable because they optimize the same LLM agent on similar datasets.

We observe the following trends. First, in terms of accuracy, `Swift` wins in 7 experiments out of 12 and is among the best two methods in 10 experiments out of 12. Although `Swift` maximizes the overall reward, it performs extremely well in all 5 reward metrics. In particular, most of its reward metrics are among the top two in 9 experiments out of 12. `Refit` performs significantly worse than `Swift` in 3 experiments: thinking OpenBookQA, standard MMLU, and standard CoSQL. Overall though, it is among the best two methods in 9 experiments out of 12. The gap from `Refit` is smaller than expected because SFT in TRL [66] is implemented with adaptive optimizers [32], which adapt to the scale of the gradient and may partially mitigate poorly scaled rewards.

The best two baselines are `STaR-GATE` and `STaR-GATE-D`. This shows the robustness of RL through SFT, the key idea in Andukuri et al. [2], which can be further improved by distillation. As discussed earlier, our work can be viewed refining this idea, where we weight the SFT update by the actual reward of the trajectory instead of an indicator of a high reward (Section 3.1). The advantage of our formulation is that it has no additional hyperparameter that decide what a high reward is, and can be properly related to the original objective (Lemma 1) and its standardization (Lemma 3). The worst baseline is `Base` and this shows the value of learning. We compare `Base` and `Refit` conversations in Appendices E and F.

**Clarifying-questions experiments.** We report our results on OpenBookQA and SciQA datasets in Tables 13 and 14. In both experiments, the accuracies of `Refit` and `Swift` are higher than those of the baselines. Although `Refit` and `Swift` do not attain the highest conversation reward metrics, they are comparable to the best baselines. Comparing to the reasoning experiments, the accuracies of answers drop significantly. This shows that the value of reasoning in our benchmarks is higher than that of asking clarifying questions.

**Ablation studies.** In Appendix B, we ablate the conversation length $n$ and logged dataset size. In addition, to alleviate the concern that our evaluation is biased due to using a single GPT-4o judge, we report results with a Claude 4 Opus judge.

Table 1: Model Performance Comparison - Thinking Mode (ARC)

| Model | Accuracy | Thinking (%) | R Overall | R Accuracy | R Reasoning | R Comprehensive | R Pedagogic | R Confidence |
|---|---|---|---|---|---|---|---|---|
| Swift (ours) | **0.7993 ± 0.0236** | **97.9 ± 0.0** | **7.19 ± 0.14** | **8.12 ± 0.17** | **7.46 ± 0.12** | **6.60 ± 0.11** | **6.95 ± 0.13** | **7.75 ± 0.17** |
| Refit (ours) | 0.7889 ± 0.0240 | **97.9 ± 0.0** | 7.12 ± 0.14 | 8.03 ± 0.17 | 7.37 ± 0.13 | 6.56 ± 0.11 | 6.88 ± 0.14 | 7.66 ± 0.18 |
| DPO | 0.6471 ± 0.0281 | 8.7 ± 0.0 | 5.72 ± 0.18 | 6.84 ± 0.22 | 6.05 ± 0.16 | 5.30 ± 0.15 | 5.21 ± 0.17 | 6.02 ± 0.21 |
| STaR-GATE | 0.6990 ± 0.0270 | 90.0 ± 0.0 | 6.67 ± 0.17 | 7.48 ± 0.20 | 6.94 ± 0.16 | 6.22 ± 0.14 | 6.50 ± 0.16 | 7.11 ± 0.21 |
| Base | 0.3772 ± 0.0146 | 75.1 ± 0.0 | 6.47 ± 0.12 | 7.32 ± 0.14 | 6.56 ± 0.11 | 5.80 ± 0.09 | 6.40 ± 0.11 | 6.92 ± 0.16 |
| STaR-GATE-D | 0.7578 ± 0.0252 | 23.9 ± 0.0 | 5.47 ± 0.16 | 6.99 ± 0.20 | 5.65 ± 0.16 | 4.83 ± 0.14 | 4.74 ± 0.16 | 5.95 ± 0.19 |
| StepDPO | 0.6401 ± 0.0282 | 8.0 ± 0.0 | 5.46 ± 0.18 | 6.60 ± 0.22 | 5.76 ± 0.17 | 5.04 ± 0.15 | 4.88 ± 0.17 | 5.83 ± 0.21 |

Table 2: Model Performance Comparison - Thinking Mode (MMLU)

| Model | Accuracy | Thinking (%) | R Overall | R Accuracy | R Reasoning | R Comprehensive | R Pedagogic | R Confidence |
|---|---|---|---|---|---|---|---|---|
| Swift (ours) | 0.7032 ± 0.0367 | 97.4 ± 0.0 | 5.59 ± 0.22 | 6.42 ± 0.26 | 5.94 ± 0.20 | 5.10 ± 0.18 | 5.23 ± 0.20 | 6.14 ± 0.26 |
| Refit (ours) | **0.7097 ± 0.0365** | **98.1 ± 0.0** | 5.59 ± 0.22 | 6.43 ± 0.26 | 5.94 ± 0.20 | 5.06 ± 0.18 | 5.19 ± 0.20 | 6.11 ± 0.26 |
| DPO | 0.6387 ± 0.0386 | 7.1 ± 0.0 | 4.77 ± 0.23 | 5.71 ± 0.29 | 5.09 ± 0.22 | 4.35 ± 0.20 | 4.24 ± 0.22 | 5.07 ± 0.28 |
| STaR-GATE | 0.6000 ± 0.0393 | 81.3 ± 0.0 | 5.34 ± 0.24 | 5.91 ± 0.29 | 5.70 ± 0.22 | 4.98 ± 0.20 | 5.15 ± 0.22 | 5.63 ± 0.29 |
| Base | 0.2774 ± 0.0127 | 53.5 ± 0.0 | **5.87 ± 0.16** | **6.57 ± 0.20** | **6.03 ± 0.15** | **5.19 ± 0.14** | **5.97 ± 0.15** | **6.19 ± 0.22** |
| STaR-GATE-D | 0.5548 ± 0.0399 | 25.2 ± 0.0 | 4.23 ± 0.23 | 4.96 ± 0.28 | 4.57 ± 0.22 | 3.93 ± 0.20 | 3.77 ± 0.21 | 4.34 ± 0.27 |
| StepDPO | 0.6387 ± 0.0386 | 5.2 ± 0.0 | 4.94 ± 0.23 | 5.88 ± 0.28 | 5.26 ± 0.21 | 4.50 ± 0.20 | 4.45 ± 0.22 | 5.31 ± 0.28 |

Table 3: Model Performance Comparison - Thinking Mode (OpenBookQA)

| Model | Accuracy | Thinking (%) | R Overall | R Accuracy | R Reasoning | R Comprehensive | R Pedagogic | R Confidence |
|---|---|---|---|---|---|---|---|---|
| Swift (ours) | 0.6814 ± 0.0310 | 96.5 ± 0.0 | **6.16 ± 0.21** | **6.86 ± 0.24** | **6.49 ± 0.19** | **5.89 ± 0.15** | **5.99 ± 0.19** | **6.52 ± 0.25** |
| Refit (ours) | 0.6504 ± 0.0317 | 96.5 ± 0.0 | 5.84 ± 0.22 | 6.63 ± 0.26 | 6.12 ± 0.21 | 5.58 ± 0.17 | 5.62 ± 0.21 | 6.25 ± 0.26 |
| DPO | 0.6195 ± 0.0323 | 10.6 ± 0.0 | 5.09 ± 0.21 | 6.21 ± 0.27 | 5.35 ± 0.20 | 4.82 ± 0.18 | 4.47 ± 0.20 | 5.55 ± 0.25 |
| STaR-GATE | 0.6549 ± 0.0316 | 92.5 ± 0.0 | 6.01 ± 0.21 | 6.68 ± 0.25 | 6.35 ± 0.20 | 5.80 ± 0.16 | 5.78 ± 0.20 | 6.36 ± 0.26 |
| Base | 0.3628 ± 0.0175 | 74.3 ± 0.0 | 5.99 ± 0.15 | 6.77 ± 0.19 | 6.15 ± 0.14 | 5.43 ± 0.12 | 5.95 ± 0.14 | 6.31 ± 0.20 |
| STaR-GATE-D | **0.6903 ± 0.0308** | 20.8 ± 0.0 | 5.21 ± 0.19 | 6.64 ± 0.25 | 5.40 ± 0.18 | 4.73 ± 0.16 | 4.35 ± 0.17 | 5.70 ± 0.23 |
| StepDPO | 0.6106 ± 0.0324 | 11.5 ± 0.0 | 4.90 ± 0.21 | 6.14 ± 0.27 | 5.06 ± 0.20 | 4.56 ± 0.18 | 4.29 ± 0.20 | 5.33 ± 0.25 |

Table 4: Model Performance Comparison - Thinking Mode (SciQA)

| Model | Accuracy | Thinking (%) | R Overall | R Accuracy | R Reasoning | R Comprehensive | R Pedagogic | R Confidence |
|---|---|---|---|---|---|---|---|---|
| Swift (ours) | **0.9248 ± 0.0175** | **99.1 ± 0.0** | 7.61 ± 0.12 | 8.84 ± 0.14 | 7.73 ± 0.11 | 6.76 ± 0.10 | 7.11 ± 0.13 | **8.45 ± 0.15** |
| Refit (ours) | 0.9159 ± 0.0185 | 96.0 ± 0.0 | **7.64 ± 0.12** | **8.87 ± 0.14** | **7.76 ± 0.11** | **6.81 ± 0.10** | **7.13 ± 0.12** | 8.43 ± 0.15 |
| DPO | 0.7920 ± 0.0270 | 5.8 ± 0.0 | 5.96 ± 0.18 | 7.61 ± 0.22 | 6.08 ± 0.18 | 5.29 ± 0.16 | 5.14 ± 0.18 | 6.50 ± 0.22 |
| STaR-GATE | 0.8186 ± 0.0256 | 90.3 ± 0.0 | 7.08 ± 0.18 | 8.17 ± 0.21 | 7.27 ± 0.16 | 6.49 ± 0.14 | 6.69 ± 0.17 | 7.69 ± 0.21 |
| Base | 0.4956 ± 0.0076 | 73.5 ± 0.0 | 7.00 ± 0.10 | 8.12 ± 0.11 | 7.03 ± 0.10 | 6.11 ± 0.09 | 6.84 ± 0.11 | 7.78 ± 0.13 |
| STaR-GATE-D | 0.9027 ± 0.0197 | 21.7 ± 0.0 | 6.58 ± 0.16 | 8.19 ± 0.18 | 6.72 ± 0.16 | 5.78 ± 0.14 | 5.73 ± 0.17 | 7.24 ± 0.18 |
| StepDPO | 0.8186 ± 0.0256 | 7.5 ± 0.0 | 6.29 ± 0.18 | 7.87 ± 0.21 | 6.36 ± 0.18 | 5.57 ± 0.16 | 5.42 ± 0.18 | 6.89 ± 0.22 |

Table 5: Model Performance Comparison - Thinking Mode (CoSQL)

| Model | Accuracy | Thinking (%) | R Overall | R Accuracy | R Reasoning | R Comprehensive | R Pedagogic | R Confidence |
|---|---|---|---|---|---|---|---|---|
| Swift (ours) | **0.6500 ± 0.0435** | 96.7 ± 0.0 | 4.87 ± 0.21 | 5.56 ± 0.27 | 5.26 ± 0.19 | 4.62 ± 0.15 | 4.23 ± 0.16 | 5.22 ± 0.29 |
| Refit (ours) | **0.6500 ± 0.0435** | **99.2 ± 0.0** | 4.91 ± 0.21 | 5.52 ± 0.27 | 5.28 ± 0.18 | 4.63 ± 0.15 | 4.22 ± 0.17 | 5.39 ± 0.31 |
| DPO | 0.5167 ± 0.0456 | 60.0 ± 0.0 | 4.34 ± 0.19 | 4.85 ± 0.25 | 4.72 ± 0.17 | 4.27 ± 0.15 | 4.00 ± 0.16 | 4.29 ± 0.28 |
| STaR-GATE | 0.6167 ± 0.0444 | 90.0 ± 0.0 | 5.28 ± 0.24 | 5.78 ± 0.30 | 5.51 ± 0.22 | **5.19 ± 0.16** | 4.90 ± 0.20 | 5.54 ± 0.33 |
| Base | 0.2000 ± 0.0143 | 65.8 ± 0.0 | **5.65 ± 0.17** | **6.17 ± 0.22** | **5.88 ± 0.15** | 5.16 ± 0.13 | **5.84 ± 0.15** | **5.87 ± 0.27** |
| STaR-GATE-D | 0.4917 ± 0.0456 | 57.5 ± 0.0 | 3.94 ± 0.17 | 4.49 ± 0.22 | 4.45 ± 0.16 | 3.89 ± 0.14 | 3.58 ± 0.15 | 3.74 ± 0.25 |
| StepDPO | 0.5250 ± 0.0456 | 60.0 ± 0.0 | 4.37 ± 0.20 | 4.82 ± 0.26 | 4.81 ± 0.18 | 4.26 ± 0.15 | 4.08 ± 0.18 | 4.38 ± 0.29 |

Table 6: Model Performance Comparison - Thinking Mode (MathDial)

| Model | Accuracy | Thinking (%) | R Overall | R Accuracy | R Reasoning | R Comprehensive | R Pedagogic | R Confidence |
|---|---|---|---|---|---|---|---|---|
| Swift (ours) | **0.1933 ± 0.0228** | 99.3 ± 0.0 | 1.88 ± 0.07 | 1.91 ± 0.07 | 2.42 ± 0.07 | 2.15 ± 0.07 | 1.83 ± 0.07 | 1.61 ± 0.09 |
| Refit (ours) | 0.0867 ± 0.0162 | **100.0 ± 0.0** | 2.38 ± 0.07 | 2.33 ± 0.07 | 3.13 ± 0.08 | 2.56 ± 0.08 | 2.43 ± 0.07 | **1.63 ± 0.07** |
| DPO | 0.1467 ± 0.0204 | 25.0 ± 0.0 | 1.61 ± 0.05 | 1.63 ± 0.06 | 2.23 ± 0.05 | 1.78 ± 0.07 | 1.56 ± 0.05 | 1.40 ± 0.06 |
| STaR-GATE | 0.0467 ± 0.0122 | **100.0 ± 0.0** | **2.46 ± 0.06** | **2.40 ± 0.07** | **3.28 ± 0.07** | **2.65 ± 0.07** | **2.45 ± 0.07** | 1.53 ± 0.05 |
| Base | 0.0000 ± 0.0212 | 87.7 ± 0.0 | 2.01 ± 0.06 | 2.28 ± 0.07 | 2.67 ± 0.07 | 1.77 ± 0.05 | 2.20 ± 0.07 | 1.39 ± 0.09 |
| STaR-GATE-D | 0.1167 ± 0.0185 | 95.0 ± 0.0 | 1.69 ± 0.06 | 1.71 ± 0.06 | 2.30 ± 0.07 | 1.81 ± 0.06 | 1.63 ± 0.06 | 1.35 ± 0.06 |
| StepDPO | 0.1467 ± 0.0204 | 25.7 ± 0.0 | 1.58 ± 0.05 | 1.61 ± 0.06 | 2.21 ± 0.06 | 1.72 ± 0.06 | 1.53 ± 0.05 | 1.40 ± 0.06 |

## 5 Related Work

We briefly review related work in three paragraphs: classic RL, RL with large language models, and supervised learning. A more detailed review is in Appendix C.

**Classic RL.** Conversation optimization using offline RL [30] is a classic topic and Section 6.6 of Levine et al. [35] reviews it in detail. Zhou et al. [81] proposed online and offline policy gradients

Table 7: Model Performance Comparison - Standard Mode (ARC)

| Model | Accuracy | Thinking (%) | R Overall | R Accuracy | R Reasoning | R Comprehensive | R Pedagogic | R Confidence |
|---|---|---|---|---|---|---|---|---|
| Swift (ours) | $\underline{0.7778 \pm 0.0289}$ | $0.0 \pm 0.0$ | $7.26 \pm 0.19$ | $8.04 \pm 0.22$ | $7.51 \pm 0.17$ | $6.76 \pm 0.14$ | $7.12 \pm 0.18$ | $7.82 \pm 0.23$ |
| Refit (ours) | $0.7729 \pm 0.0291$ | $0.0 \pm 0.0$ | $7.23 \pm 0.19$ | $7.98 \pm 0.22$ | $7.44 \pm 0.18$ | $\underline{6.80 \pm 0.14}$ | $7.03 \pm 0.18$ | $7.66 \pm 0.23$ |
| DPO | $0.6377 \pm 0.0334$ | $0.0 \pm 0.0$ | $5.68 \pm 0.20$ | $6.51 \pm 0.25$ | $6.06 \pm 0.18$ | $5.41 \pm 0.16$ | $5.26 \pm 0.19$ | $5.78 \pm 0.25$ |
| STaR-GATE | $\mathbf{0.7971 \pm 0.0280}$ | $0.0 \pm 0.0$ | $\mathbf{7.49 \pm 0.18}$ | $\mathbf{8.25 \pm 0.21}$ | $\mathbf{7.67 \pm 0.17}$ | $\mathbf{6.93 \pm 0.14}$ | $\mathbf{7.36 \pm 0.17}$ | $\mathbf{8.02 \pm 0.22}$ |
| Base | $0.5652 \pm 0.0142$ | $0.0 \pm 0.0$ | $6.87 \pm 0.14$ | $7.68 \pm 0.18$ | $6.97 \pm 0.13$ | $6.25 \pm 0.11$ | $6.75 \pm 0.14$ | $7.21 \pm 0.20$ |
| STaR-GATE-D | $0.7101 \pm 0.0315$ | $0.0 \pm 0.0$ | $5.95 \pm 0.18$ | $6.96 \pm 0.22$ | $6.29 \pm 0.17$ | $5.56 \pm 0.14$ | $5.42 \pm 0.17$ | $6.18 \pm 0.22$ |
| StepDPO | $0.6280 \pm 0.0336$ | $0.0 \pm 0.0$ | $5.76 \pm 0.20$ | $6.55 \pm 0.25$ | $6.19 \pm 0.18$ | $5.54 \pm 0.15$ | $5.43 \pm 0.19$ | $5.84 \pm 0.25$ |

Table 8: Model Performance Comparison - Standard Mode (MMLU)

| Model | Accuracy | Thinking (%) | R Overall | R Accuracy | R Reasoning | R Comprehensive | R Pedagogic | R Confidence |
|---|---|---|---|---|---|---|---|---|
| Swift (ours) | $\mathbf{0.7218 \pm 0.0389}$ | $0.0 \pm 0.0$ | $\mathbf{6.08 \pm 0.25}$ | $\mathbf{6.88 \pm 0.29}$ | $\mathbf{6.32 \pm 0.23}$ | $\mathbf{5.50 \pm 0.21}$ | $5.80 \pm 0.23$ | $\mathbf{6.71 \pm 0.30}$ |
| Refit (ours) | $\underline{0.6917 \pm 0.0400}$ | $0.0 \pm 0.0$ | $5.93 \pm 0.26$ | $6.72 \pm 0.31$ | $6.23 \pm 0.24$ | $5.42 \pm 0.21$ | $\underline{5.56 \pm 0.25}$ | $6.36 \pm 0.31$ |
| DPO | $0.5489 \pm 0.0431$ | $0.0 \pm 0.0$ | $4.86 \pm 0.25$ | $5.52 \pm 0.31$ | $5.30 \pm 0.23$ | $4.61 \pm 0.21$ | $4.56 \pm 0.23$ | $4.92 \pm 0.30$ |
| STaR-GATE | $0.6842 \pm 0.0403$ | $0.0 \pm 0.0$ | $5.93 \pm 0.26$ | $6.68 \pm 0.31$ | $6.20 \pm 0.25$ | $5.41 \pm 0.22$ | $5.59 \pm 0.25$ | $6.41 \pm 0.31$ |
| Base | $0.3008 \pm 0.0165$ | $0.0 \pm 0.0$ | $\underline{5.97 \pm 0.19}$ | $6.74 \pm 0.23$ | $6.16 \pm 0.18$ | $5.32 \pm 0.16$ | $\mathbf{5.95 \pm 0.18}$ | $6.11 \pm 0.26$ |
| STaR-GATE-D | $0.5940 \pm 0.0426$ | $0.0 \pm 0.0$ | $4.98 \pm 0.25$ | $5.75 \pm 0.30$ | $5.29 \pm 0.24$ | $4.65 \pm 0.21$ | $4.53 \pm 0.24$ | $5.26 \pm 0.31$ |
| StepDPO | $0.5263 \pm 0.0433$ | $0.0 \pm 0.0$ | $4.77 \pm 0.26$ | $5.44 \pm 0.32$ | $5.17 \pm 0.25$ | $4.49 \pm 0.22$ | $4.38 \pm 0.23$ | $4.94 \pm 0.31$ |

Table 9: Model Performance Comparison - Standard Mode (OpenBookQA)

| Model | Accuracy | Thinking (%) | R Overall | R Accuracy | R Reasoning | R Comprehensive | R Pedagogic | R Confidence |
|---|---|---|---|---|---|---|---|---|
| Swift (ours) | $\mathbf{0.7662 \pm 0.0299}$ | $0.0 \pm 0.0$ | $\mathbf{6.85 \pm 0.21}$ | $\mathbf{7.73 \pm 0.24}$ | $\mathbf{7.09 \pm 0.19}$ | $\mathbf{6.42 \pm 0.15}$ | $\mathbf{6.59 \pm 0.20}$ | $\mathbf{7.45 \pm 0.25}$ |
| Refit (ours) | $\underline{0.7562 \pm 0.0303}$ | $0.0 \pm 0.0$ | $\underline{6.73 \pm 0.21}$ | $7.66 \pm 0.25$ | $6.96 \pm 0.20$ | $6.29 \pm 0.16$ | $6.43 \pm 0.21$ | $7.25 \pm 0.25$ |
| DPO | $0.5025 \pm 0.0353$ | $0.0 \pm 0.0$ | $4.95 \pm 0.21$ | $5.49 \pm 0.26$ | $5.43 \pm 0.19$ | $5.04 \pm 0.16$ | $4.66 \pm 0.20$ | $4.95 \pm 0.26$ |
| STaR-GATE | $0.7512 \pm 0.0305$ | $0.0 \pm 0.0$ | $6.69 \pm 0.22$ | $7.54 \pm 0.25$ | $6.96 \pm 0.20$ | $6.27 \pm 0.16$ | $\underline{6.50 \pm 0.21}$ | $7.23 \pm 0.26$ |
| Base | $0.4328 \pm 0.0180$ | $0.0 \pm 0.0$ | $6.22 \pm 0.16$ | $6.95 \pm 0.21$ | $6.37 \pm 0.15$ | $5.65 \pm 0.13$ | $6.12 \pm 0.15$ | $6.51 \pm 0.21$ |
| STaR-GATE-D | $0.7114 \pm 0.0320$ | $0.0 \pm 0.0$ | $5.84 \pm 0.19$ | $6.96 \pm 0.24$ | $6.21 \pm 0.18$ | $5.52 \pm 0.15$ | $5.24 \pm 0.18$ | $6.21 \pm 0.23$ |
| StepDPO | $0.5174 \pm 0.0352$ | $0.0 \pm 0.0$ | $4.92 \pm 0.22$ | $5.54 \pm 0.27$ | $5.36 \pm 0.20$ | $4.97 \pm 0.17$ | $4.65 \pm 0.20$ | $4.97 \pm 0.27$ |

Table 10: Model Performance Comparison - Standard Mode (SciQA)

| Model | Accuracy | Thinking (%) | R Overall | R Accuracy | R Reasoning | R Comprehensive | R Pedagogic | R Confidence |
|---|---|---|---|---|---|---|---|---|
| Swift (ours) | $\mathbf{0.9502 \pm 0.0153}$ | $0.0 \pm 0.0$ | $\mathbf{8.04 \pm 0.12}$ | $\mathbf{9.13 \pm 0.13}$ | $\mathbf{8.12 \pm 0.11}$ | $\underline{7.17 \pm 0.10}$ | $\mathbf{7.71 \pm 0.13}$ | $\mathbf{8.88 \pm 0.15}$ |
| Refit (ours) | $\underline{0.9453 \pm 0.0160}$ | $0.0 \pm 0.0$ | $\mathbf{8.04 \pm 0.12}$ | $9.08 \pm 0.14$ | $8.11 \pm 0.11$ | $\mathbf{7.20 \pm 0.10}$ | $\underline{7.69 \pm 0.13}$ | $8.87 \pm 0.15$ |
| DPO | $0.7612 \pm 0.0301$ | $0.0 \pm 0.0$ | $6.41 \pm 0.19$ | $7.44 \pm 0.23$ | $6.72 \pm 0.17$ | $6.00 \pm 0.15$ | $6.02 \pm 0.19$ | $6.78 \pm 0.23$ |
| STaR-GATE | $0.9005 \pm 0.0211$ | $0.0 \pm 0.0$ | $7.85 \pm 0.16$ | $8.88 \pm 0.18$ | $7.98 \pm 0.14$ | $7.06 \pm 0.13$ | $7.52 \pm 0.16$ | $8.62 \pm 0.19$ |
| Base | $0.6517 \pm 0.0086$ | $0.0 \pm 0.0$ | $7.48 \pm 0.10$ | $8.56 \pm 0.11$ | $7.52 \pm 0.10$ | $6.55 \pm 0.09$ | $7.34 \pm 0.10$ | $8.10 \pm 0.13$ |
| STaR-GATE-D | $0.9005 \pm 0.0211$ | $0.0 \pm 0.0$ | $6.90 \pm 0.15$ | $8.39 \pm 0.17$ | $7.13 \pm 0.14$ | $6.20 \pm 0.13$ | $6.03 \pm 0.16$ | $7.42 \pm 0.18$ |
| StepDPO | $0.7463 \pm 0.0307$ | $0.0 \pm 0.0$ | $6.23 \pm 0.20$ | $7.25 \pm 0.24$ | $6.52 \pm 0.19$ | $5.88 \pm 0.16$ | $5.78 \pm 0.19$ | $6.53 \pm 0.25$ |

Table 11: Model Performance Comparison - Standard Mode (CoSQL)

| Model | Accuracy | Thinking (%) | R Overall | R Accuracy | R Reasoning | R Comprehensive | R Pedagogic | R Confidence |
|---|---|---|---|---|---|---|---|---|
| Swift (ours) | $\mathbf{0.6583 \pm 0.0433}$ | $0.0 \pm 0.0$ | $\mathbf{5.45 \pm 0.24}$ | $\mathbf{5.97 \pm 0.30}$ | $\mathbf{5.72 \pm 0.22}$ | $\mathbf{5.28 \pm 0.17}$ | $4.95 \pm 0.21$ | $\mathbf{5.85 \pm 0.33}$ |
| Refit (ours) | $\underline{0.6250 \pm 0.0442}$ | $0.0 \pm 0.0$ | $5.16 \pm 0.24$ | $5.64 \pm 0.30$ | $5.47 \pm 0.22$ | $4.99 \pm 0.18$ | $4.72 \pm 0.20$ | $5.52 \pm 0.32$ |
| DPO | $0.2833 \pm 0.0411$ | $0.0 \pm 0.0$ | $4.19 \pm 0.20$ | $4.37 \pm 0.25$ | $4.79 \pm 0.19$ | $4.49 \pm 0.15$ | $4.13 \pm 0.17$ | $3.69 \pm 0.27$ |
| STaR-GATE | $0.6083 \pm 0.0446$ | $0.0 \pm 0.0$ | $5.34 \pm 0.25$ | $5.81 \pm 0.31$ | $5.65 \pm 0.23$ | $\underline{5.26 \pm 0.18}$ | $4.99 \pm 0.22$ | $\underline{5.57 \pm 0.34}$ |
| Base | $0.1250 \pm 0.0117$ | $0.0 \pm 0.0$ | $\underline{5.38 \pm 0.16}$ | $\underline{5.88 \pm 0.22}$ | $\underline{5.66 \pm 0.15}$ | $4.92 \pm 0.12$ | $\mathbf{5.49 \pm 0.14}$ | $5.13 \pm 0.24$ |
| STaR-GATE-D | $0.2083 \pm 0.0371$ | $0.0 \pm 0.0$ | $3.62 \pm 0.19$ | $3.82 \pm 0.23$ | $4.25 \pm 0.18$ | $4.02 \pm 0.16$ | $3.73 \pm 0.17$ | $3.03 \pm 0.26$ |
| StepDPO | $0.2917 \pm 0.0415$ | $0.0 \pm 0.0$ | $4.21 \pm 0.20$ | $4.45 \pm 0.26$ | $4.85 \pm 0.18$ | $4.50 \pm 0.15$ | $4.10 \pm 0.17$ | $3.73 \pm 0.28$ |

Table 12: Model Performance Comparison - Standard Mode (MathDial)

| Model | Accuracy | Thinking (%) | R Overall | R Accuracy | R Reasoning | R Comprehensive | R Pedagogic | R Confidence |
|---|---|---|---|---|---|---|---|---|
| Swift (ours) | $0.0967 \pm 0.0171$ | $0.0 \pm 0.0$ | $\underline{2.43 \pm 0.07}$ | $2.41 \pm 0.07$ | $3.09 \pm 0.08$ | $\mathbf{2.68 \pm 0.07}$ | $\mathbf{2.45 \pm 0.07}$ | $\mathbf{1.66 \pm 0.07}$ |
| Refit (ours) | $0.0600 \pm 0.0137$ | $0.0 \pm 0.0$ | $\mathbf{2.43 \pm 0.07}$ | $\mathbf{2.50 \pm 0.08}$ | $\mathbf{3.15 \pm 0.08}$ | $2.68 \pm 0.07$ | $2.41 \pm 0.08$ | $1.63 \pm 0.07$ |
| DPO | $\mathbf{0.2100 \pm 0.0235}$ | $0.0 \pm 0.0$ | $1.85 \pm 0.06$ | $1.90 \pm 0.07$ | $2.41 \pm 0.06$ | $2.00 \pm 0.07$ | $1.77 \pm 0.06$ | $1.58 \pm 0.07$ |
| STaR-GATE | $0.1067 \pm 0.0178$ | $0.0 \pm 0.0$ | $2.29 \pm 0.07$ | $2.25 \pm 0.07$ | $2.94 \pm 0.08$ | $2.58 \pm 0.08$ | $2.30 \pm 0.07$ | $1.56 \pm 0.07$ |
| Base | $0.0000 \pm 0.0168$ | $0.0 \pm 0.0$ | $1.90 \pm 0.06$ | $2.20 \pm 0.07$ | $2.54 \pm 0.07$ | $1.81 \pm 0.05$ | $2.01 \pm 0.07$ | $1.20 \pm 0.08$ |
| STaR-GATE-D | $0.2000 \pm 0.0231$ | $0.0 \pm 0.0$ | $1.55 \pm 0.04$ | $1.63 \pm 0.05$ | $1.95 \pm 0.05$ | $1.79 \pm 0.06$ | $1.51 \pm 0.04$ | $1.31 \pm 0.05$ |
| StepDPO | $\underline{0.2067 \pm 0.0234}$ | $0.0 \pm 0.0$ | $1.86 \pm 0.06$ | $1.87 \pm 0.06$ | $2.43 \pm 0.06$ | $2.03 \pm 0.07$ | $1.78 \pm 0.06$ | $1.55 \pm 0.06$ |

for improving language quality. Neither this approach nor other classic techniques, like Q-learning [70, 44], can be directly applied to LLMs. Peters and Schaal [49] formulated RL as reward-weighted regression and proposed an EM algorithm for solving it, where an auxiliary reweighting distribution is optimized together with the policy. In contrast, Refit and Swift are policy gradient algorithms that do not require any auxiliary distribution. Peng et al. [48] proposed maximizing the log-probability of actions weighted by an exponentiated advantage. Refit and Swift can also be viewed as behavioral

Table 13: Model Performance Comparison - Thinking Mode (OpenBookQA Clarifying Questions)

| Model | Accuracy | Thinking (%) | R Overall | R Accuracy | R Reasoning | R Comprehensive | R Pedagogic | R Confidence |
|---|---|---|---|---|---|---|---|---|
| Swift (ours) | $0.2400 \pm 0.0604$ | $46.0 \pm 7.0$ | $4.67 \pm 0.21$ | $5.70 \pm 0.30$ | $5.39 \pm 0.22$ | $4.41 \pm 0.19$ | $5.13 \pm 0.21$ | $5.23 \pm 0.19$ |
| Refit (ours) | $\mathbf{0.2800 \pm 0.0635}$ | $26.0 \pm 6.2$ | $4.55 \pm 0.21$ | $5.55 \pm 0.32$ | $5.32 \pm 0.23$ | $4.37 \pm 0.19$ | $4.95 \pm 0.22$ | $4.94 \pm 0.19$ |
| Base | $0.1000 \pm 0.0424$ | $\mathbf{47.0 \pm 4.9}$ | $\mathbf{4.89 \pm 0.24}$ | $\mathbf{5.93 \pm 0.31}$ | $\mathbf{5.63 \pm 0.22}$ | $\mathbf{4.60 \pm 0.20}$ | $\mathbf{5.37 \pm 0.23}$ | $\mathbf{5.35 \pm 0.25}$ |
| DPO | $0.1000 \pm 0.0424$ | $30.0 \pm 6.5$ | $4.27 \pm 0.24$ | $5.29 \pm 0.32$ | $5.05 \pm 0.25$ | $4.04 \pm 0.22$ | $4.75 \pm 0.24$ | $4.87 \pm 0.21$ |
| STaR-GATE-D | $0.1000 \pm 0.0424$ | $4.0 \pm 2.8$ | $4.11 \pm 0.21$ | $5.13 \pm 0.30$ | $4.85 \pm 0.22$ | $3.81 \pm 0.20$ | $4.42 \pm 0.21$ | $4.59 \pm 0.20$ |
| StepDPO | $0.2000 \pm 0.0566$ | $26.0 \pm 6.2$ | $4.27 \pm 0.26$ | $5.31 \pm 0.35$ | $5.01 \pm 0.28$ | $4.07 \pm 0.24$ | $4.67 \pm 0.26$ | $4.73 \pm 0.25$ |

Table 14: Model Performance Comparison - Thinking Mode (SciQA Clarifying Questions)

| Model | Accuracy | Thinking (%) | R Overall | R Accuracy | R Reasoning | R Comprehensive | R Pedagogic | R Confidence |
|---|---|---|---|---|---|---|---|---|
| Swift (ours) | $\mathbf{0.2600 \pm 0.0620}$ | $62.0 \pm 6.9$ | $4.63 \pm 0.24$ | $5.77 \pm 0.33$ | $5.39 \pm 0.24$ | $4.44 \pm 0.22$ | $5.07 \pm 0.23$ | $4.96 \pm 0.24$ |
| Refit (ours) | $0.2400 \pm 0.0604$ | $68.0 \pm 6.6$ | $5.03 \pm 0.25$ | $6.21 \pm 0.31$ | $5.81 \pm 0.24$ | $4.75 \pm 0.20$ | $5.42 \pm 0.23$ | $5.33 \pm 0.22$ |
| Base | $0.0600 \pm 0.0336$ | $\mathbf{86.0 \pm 4.9}$ | $\mathbf{5.47 \pm 0.23}$ | $\mathbf{6.69 \pm 0.28}$ | $\mathbf{6.14 \pm 0.21}$ | $\mathbf{5.10 \pm 0.20}$ | $\mathbf{5.95 \pm 0.21}$ | $\mathbf{5.87 \pm 0.21}$ |
| DPO | $0.0800 \pm 0.0384$ | $24.0 \pm 6.0$ | $4.44 \pm 0.25$ | $5.53 \pm 0.33$ | $5.17 \pm 0.26$ | $4.19 \pm 0.22$ | $4.86 \pm 0.26$ | $4.86 \pm 0.26$ |
| STaR-GATE-D | $0.1400 \pm 0.0491$ | $16.0 \pm 5.2$ | $4.01 \pm 0.22$ | $4.95 \pm 0.30$ | $4.76 \pm 0.24$ | $3.67 \pm 0.21$ | $4.30 \pm 0.23$ | $4.65 \pm 0.21$ |
| StepDPO | $0.0800 \pm 0.0384$ | $34.0 \pm 6.7$ | $4.70 \pm 0.23$ | $5.85 \pm 0.32$ | $5.36 \pm 0.25$ | $4.44 \pm 0.21$ | $5.09 \pm 0.23$ | $5.13 \pm 0.24$ |

cloning [50, 28] where the rewards and advantages, respectively, weigh the logged trajectories by their importance for learning.

**RL with LLMs.** The closest related works are Andukuri et al. [2] and Chen et al. [11], both of which used RL to learn clarifying questions from simulated conversations. Andukuri et al. [2] fine-tuned on most rewarding trajectories. Chen et al. [11] generated alternative responses for each step of the conversation and then optimized for better responses using DPO [52]. The main difference in our work is that we directly optimize for rewards. Our work is also broadly related to LLM planning: Huang et al. [27] planned with pre-trained models, Hao et al. [21] used Monte Carlo tree search to search for policies, and Wang et al. [69] re-planned interactively based on reached sub-goals.

**Supervised learning.** Many works have focused on clarifying user prompts by asking clarifying questions [39, 75]. Zelikman et al. [75] proposed a simple yet powerful approach: learning from rationales for successes and corrected failures. The problem of whether to ask a clarifying question has been studied extensively [40, 8, 36], giving rise to new benchmarks [8, 78] and surveys [45, 77]. These studies have also been extended to vision-language models [20, 65, 10]. In comparison, we take an RL approach.

# 6 Conclusions

Offline RL is a variant of reinforcement learning where the policy is optimized over a previously collected dataset of trajectories and rewards. In our work, we propose a practical approach to offline RL with large language models. The key idea is to recast RL as reward-weighted fine-tuning, which can be implemented using similar techniques to SFT. We also propose an algorithm for standardized rewards, which can be more statistically efficient in practice. To show the value of our approach, we apply it to learning multi-turn QA policies, where the agent reasons about potential answers or asks clarifying questions. Our work stands in a stark contrast to state-of-the-art methods in this domain, based on SFT and DPO, which have additional hyper-parameters and do not directly optimize for rewards. We compare to these works empirically, and report major gains in both optimized rewards and language quality.

**Limitations.** The computational cost of RL tends to be much higher than that of supervised learning. We address this issue partially by proposing a reduction of offline RL to SFT, which is a supervised learning technique. In addition, the quality of the logged dataset is critical for offline RL. We do not focus on this aspect of the problem and instead rely on a common method to obtain a diverse dataset: simulate conversation trajectories using different temperatures in the LLM. Finally, similarly to the closest related works [2, 11], we do not conduct a human evaluation. To alleviate the concern that our evaluation is biased due to using a single GPT-4o judge, we report results with a Claude 4 Opus judge in Appendix B.

**Future work.** We note that our proposed algorithms Refit and Swift are general, and therefore can be applied to other domains than QA. We focused on QA due to many established benchmarks and baselines in this domain, which allow us to showcase the benefit of directly optimizing rewards.

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

# A Proofs and Supporting Lemmas

This section contains proofs of our main claims and supporting lemmas.

## A.1 Proof of Lemma 2

We first note that

$$\mathbb{E}_{x\sim q,\, \tau_n \sim \pi(\cdot|x;\theta)}\left[\tilde{r}(x,\tau_n)\right] = \mathbb{E}_{x\sim q}\left[\frac{1}{\sigma(x)}\mathbb{E}_{\tau_n \sim \pi(\cdot|x;\theta)}\left[r(x,\tau_n)\,|\,x\right]\right] - C\,,$$

where $C = \mathbb{E}_{x\sim q}\left[\mu(x)/\sigma(x)\right]$ is a constant independent of $\theta$. Since all $\mathbb{E}_{\tau_n \sim \pi(\cdot|x;\theta)}\left[r(x,\tau_n)\,|\,x\right]$ are jointly maximized by $\theta_*$ and the weights $1/\sigma(x)$ are non-negative, $\theta_*$ also maximizes any weighted combination of the objectives. This completes our proof.

## A.2 Proof of Lemma 3

Using basic algebra,

$$\begin{aligned}
\mathbb{E}_{x\sim q,\, \tau_n \sim \pi(\cdot|x;\theta)}\left[\tilde{r}(x,\tau_n)\right] &= \mathbb{E}_{x\sim q,\, \tau_n \sim \pi_0(\cdot|x)}\left[\tilde{r}(x,\tau_n)\frac{\pi(\tau_n\,|\,x;\theta)}{\pi_0(\tau_n\,|\,x)}\right] \\
&= \mathbb{E}_{x\sim q,\, \tau_n \sim \pi_0(\cdot|x)}\left[\tilde{r}(x,\tau_n)\left(1 + \log\frac{\pi(\tau_n\,|\,x;\theta)}{\pi_0(\tau_n\,|\,x)}\right)\right] + \Delta(\theta) \\
&= \mathbb{E}_{x\sim q,\, \tau_n \sim \pi_0(\cdot|x)}\left[\tilde{r}(x,\tau_n)\log\pi(\tau_n\,|\,x;\theta)\right] + \Delta(\theta) + C_1\,,
\end{aligned}$$

where

$$\Delta(\theta) = \mathbb{E}_{x\sim q,\, \tau_n \sim \pi_0(\cdot|x)}\left[\tilde{r}(x,\tau_n)\left(\frac{\pi(\tau_n\,|\,x;\theta)}{\pi_0(\tau_n\,|\,x)} - \left(1 + \log\frac{\pi(\tau_n\,|\,x;\theta)}{\pi_0(\tau_n\,|\,x)}\right)\right)\right]$$

and $C_1$ is a constant independent of $\theta$ defined in Lemma 1. Now we rearrange the equality, take the absolute value of both sides, and get

$$\begin{aligned}
\left|\mathbb{E}_{x\sim q,\, \tau_n \sim \pi(\cdot|x;\theta)}\left[\tilde{r}(x,\tau_n)\right] - \mathbb{E}_{x\sim q,\, \tau_n \sim \pi_0(\cdot|x)}\left[\tilde{r}(x,\tau_n)\log\pi(\tau_n\,|\,x;\theta)\right]\right| &= |C_1 + \Delta(\theta)| \\
&\leq |C_1| + |\Delta(\theta)|\,.
\end{aligned}$$

We bound $|\Delta(\theta)|$ as

$$\begin{aligned}
|\Delta(\theta)| &\leq \mathbb{E}_{x\sim q,\, \tau_n \sim \pi_0(\cdot|x)}\left[\left|\tilde{r}(x,\tau_n)\left(\frac{\pi(\tau_n\,|\,x;\theta)}{\pi_0(\tau_n\,|\,x)} - \left(1 + \log\frac{\pi(\tau_n\,|\,x;\theta)}{\pi_0(\tau_n\,|\,x)}\right)\right)\right|\right] \\
&\leq \max_{x,\tau_n}\left|\tilde{r}(x,\tau_n)\left(\frac{\pi(\tau_n\,|\,x;\theta)}{\pi_0(\tau_n\,|\,x)} - \left(1 + \log\frac{\pi(\tau_n\,|\,x;\theta)}{\pi_0(\tau_n\,|\,x)}\right)\right)\right| \\
&\leq b\max_{x,\tau_n}\left(\frac{\pi(\tau_n\,|\,x;\theta)}{\pi_0(\tau_n\,|\,x)} - \left(1 + \log\frac{\pi(\tau_n\,|\,x;\theta)}{\pi_0(\tau_n\,|\,x)}\right)\right)\,.
\end{aligned}$$

The last step holds because the rewards are in $[-b, b]$ and $u \geq 1 + \log u$. Finally, to bound $|\Delta(\theta)|$, we maximize over $\theta$. This completes the proof.

## A.3 Improvements in `Swift` Objective

Our analysis builds on Sections 1 and 2 in the note of Jin et al. [31]. Take (7) and let

$$g(x,\tau_n;\theta) = \sum_{t=1}^{n}\nabla\log\pi(a_t\,|\,x,\tau_{t-1};\theta)$$

be the random gradient inside the expectation, for random $x$ and $\tau_n$. Suppose that

$$\|g(x,\tau_n;\theta)\|_2 \leq \sigma$$

holds for any $x$, $\tau_n$, and $\theta$. Then $g(x,\tau_n;\theta)$ is a $\sigma$-norm-sub-Gaussian vector (Lemma 1 in the note). Let all rewards be non-negative (Section 2) and $r_{\max} = \max_{x,\tau_n} r(x,\tau_n)$ be the maximum reward.

Table 15: Model Performance Comparison - Thinking Mode (OpenBookQA $n = 4$)

| Model | Accuracy | Thinking (%) | R Overall | R Accuracy | R Reasoning | R Comprehensive | R Pedagogic | R Confidence |
|---|---|---|---|---|---|---|---|---|
| Swift (ours) | $\mathbf{0.9333 \pm 0.0644}$ | $\mathbf{93.3 \pm 0.0}$ | $\mathbf{8.32 \pm 0.27}$ | $\mathbf{9.27 \pm 0.38}$ | $\mathbf{8.47 \pm 0.22}$ | $\mathbf{7.40 \pm 0.24}$ | $\underline{7.80 \pm 0.35}$ | $\mathbf{9.13 \pm 0.41}$ |
| Refit (ours) | $\mathbf{0.9333 \pm 0.0644}$ | $\mathbf{93.3 \pm 0.0}$ | $\underline{8.23 \pm 0.33}$ | $\mathbf{9.27 \pm 0.37}$ | $\mathbf{8.47 \pm 0.27}$ | $\mathbf{7.40 \pm 0.31}$ | $\mathbf{7.87 \pm 0.31}$ | $\underline{8.87 \pm 0.52}$ |

Table 16: Model Performance Comparison - Thinking Mode (OpenBookQA $n = 6$)

| Model | Accuracy | Thinking (%) | R Overall | R Accuracy | R Reasoning | R Comprehensive | R Pedagogic | R Confidence |
|---|---|---|---|---|---|---|---|---|
| Swift (ours) | $0.8667 \pm 0.0878$ | $\mathbf{100.0 \pm 0.0}$ | $\underline{7.68 \pm 0.63}$ | $\underline{8.67 \pm 0.77}$ | $\underline{7.93 \pm 0.52}$ | $\underline{6.93 \pm 0.52}$ | $\underline{7.33 \pm 0.61}$ | $\underline{8.33 \pm 0.75}$ |
| Refit (ours) | $\mathbf{1.0000 \pm 0.0000}$ | $\mathbf{100.0 \pm 0.0}$ | $\mathbf{8.53 \pm 0.22}$ | $\mathbf{9.80 \pm 0.20}$ | $\mathbf{8.53 \pm 0.22}$ | $\mathbf{7.47 \pm 0.24}$ | $\mathbf{8.40 \pm 0.25}$ | $\mathbf{9.60 \pm 0.16}$ |

Table 17: Model Performance Comparison - Thinking Mode (OpenBookQA $n = 8$)

| Model | Accuracy | Thinking (%) | R Overall | R Accuracy | R Reasoning | R Comprehensive | R Pedagogic | R Confidence |
|---|---|---|---|---|---|---|---|---|
| Swift (ours) | $0.8000 \pm 0.1033$ | $73.3 \pm 0.0$ | $6.87 \pm 1.00$ | $7.87 \pm 1.06$ | $6.80 \pm 1.05$ | $6.13 \pm 0.96$ | $6.47 \pm 1.03$ | $7.47 \pm 1.05$ |
| Refit (ours) | $\mathbf{0.9333 \pm 0.0644}$ | $\mathbf{93.3 \pm 0.0}$ | $\mathbf{7.91 \pm 0.66}$ | $\mathbf{8.93 \pm 0.70}$ | $\mathbf{8.00 \pm 0.68}$ | $\mathbf{7.20 \pm 0.59}$ | $\mathbf{7.60 \pm 0.63}$ | $\mathbf{8.60 \pm 0.75}$ |

Table 18: Model Performance Comparison - Thinking Mode (OpenBookQA $n = 10$)

| Model | Accuracy | Thinking (%) | R Overall | R Accuracy | R Reasoning | R Comprehensive | R Pedagogic | R Confidence |
|---|---|---|---|---|---|---|---|---|
| Swift (ours) | $0.6000 \pm 0.1265$ | $80.0 \pm 0.0$ | $6.14 \pm 0.95$ | $7.02 \pm 1.11$ | $6.33 \pm 0.90$ | $5.53 \pm 0.80$ | $6.07 \pm 0.91$ | $6.69 \pm 1.11$ |
| Refit (ours) | $\mathbf{0.8667 \pm 0.0878}$ | $\mathbf{93.3 \pm 0.0}$ | $\mathbf{7.85 \pm 0.73}$ | $\mathbf{8.73 \pm 0.82}$ | $\mathbf{7.87 \pm 0.74}$ | $\mathbf{7.07 \pm 0.61}$ | $\mathbf{7.73 \pm 0.68}$ | $\mathbf{8.47 \pm 0.88}$ |

Then $r(x, \tau_n)g(x, \tau_n; \theta)$ is $(r_{\max}\sigma)$-norm-sub-Gaussian. The average of such vectors concentrates at (7) in the $L_2$-norm proportionally to their sub-Gaussianity parameter $r_{\max}\sigma$ (Definition 3 in the note).

Let $\mu_x = \mathbb{E}\left[r(x, \tau_n) \mid x\right]$. Since the rewards are non-negative, $|r(x, \tau_n) - \mu_x| \leq r_{\max}$ for any $x$ and $\tau_n$. Therefore, $(r(x, \tau_n) - \mu_x)g(x, \tau_n; \theta)$ is at most $(r_{\max}\sigma)$-norm-sub-Gaussian, and the average of such vectors concentrates at least as fast as without subtracting the mean. This results in a higher statistical efficiency in estimating the gradient. We note that the new estimator is biased, since

$$\mathbb{E}\left[r(x, \tau_n)g(x, \tau_n; \theta)\right] = \mathbb{E}\left[(r(x, \tau_n) - \mu_x)g(x, \tau_n; \theta)\right]$$

holds only under the assumption that $\tau_n \sim \pi(\cdot \mid x; \theta)$.

To the best of our knowledge, the normalization by $\sigma_x^2 = \text{var}\left[r(x, \tau_n) \mid x\right]$ in

$$\mathbb{E}\left[\frac{r(x, \tau_n) - \mu_x}{\sigma_x} g(x, \tau_n; \theta)\right]$$

is hard to analyze because it changes the gradient. It tends to help in practice because it renormalizes the variances of rewards across all contexts $x$. Therefore, the learned policy improves uniformly in all contexts without tuning the learning rate per context. This was popularized by GRPO [58].

## B   Ablation Studies

We ablate the performance of Swift and Refit as a function of the conversation length, for $n \in \{4, 6, 8, 10\}$, in Tables 15-18. We observe that longer conversations, corresponding to larger values of $n$, lead to lower accuracy because the task becomes harder.

To alleviate the concern that our evaluation is biased due to using a single GPT-4o judge, we report results with a Claude 4 Opus judge on ARC dataset in Table 19. The prompt is the same as in the GPT-4o judge. The best two methods are the same as in Table 1: Swift and Refit. As in Table 1, Swift and Refit attain much higher language quality scores than all baselines.

The run times of Refit and Swift are linear in sample size, because both methods make a single pass over the logged dataset. We expect the reward to increase as the number of training trajectories increases. To show this, we conduct a dataset size ablation study on ARC dataset, in both thinking and standard modes. The results for Refit and two different sample sizes are reported in Tables 20 and 21. The accuracy and language quality metrics improve with more training data.

Table 19: Claude 4 Opus Judge - Thinking Mode (ARC)

| Model | Accuracy | Thinking (%) | R Overall | R Accuracy | R Reasoning | R Comprehensive | R Pedagogic | R Confidence |
|---|---|---|---|---|---|---|---|---|
| Swift (ours) | $0.8667 \pm 0.0310$ | $97.5 \pm 0.0$ | $6.28 \pm 0.19$ | $8.76 \pm 0.26$ | $6.73 \pm 0.18$ | $5.79 \pm 0.14$ | $6.33 \pm 0.19$ | $7.24 \pm 0.23$ |
| Refit (ours) | $0.8583 \pm 0.0318$ | $98.3 \pm 0.0$ | $6.31 \pm 0.20$ | $8.69 \pm 0.27$ | $6.73 \pm 0.19$ | $5.78 \pm 0.15$ | $6.34 \pm 0.19$ | $7.25 \pm 0.23$ |
| DPO | $0.7167 \pm 0.0411$ | $7.5 \pm 0.0$ | $4.29 \pm 0.24$ | $7.28 \pm 0.37$ | $4.50 \pm 0.24$ | $3.27 \pm 0.18$ | $3.30 \pm 0.22$ | $4.92 \pm 0.28$ |
| STaR-GATE | $0.6990 \pm 0.0270$ | $90.0 \pm 0.0$ | $6.17 \pm 0.17$ | $7.48 \pm 0.20$ | $5.94 \pm 0.16$ | $5.22 \pm 0.14$ | $5.50 \pm 0.16$ | $7.11 \pm 0.21$ |
| Base | $0.3772 \pm 0.0146$ | $75.1 \pm 0.0$ | $5.47 \pm 0.12$ | $7.32 \pm 0.14$ | $5.56 \pm 0.11$ | $5.80 \pm 0.09$ | $5.40 \pm 0.11$ | $5.92 \pm 0.16$ |
| STaR-GATE-D | $0.8417 \pm 0.0333$ | $28.3 \pm 0.0$ | $4.22 \pm 0.23$ | $8.30 \pm 0.33$ | $4.17 \pm 0.24$ | $2.98 \pm 0.18$ | $3.06 \pm 0.22$ | $5.36 \pm 0.28$ |
| StepDPO | $0.7167 \pm 0.0411$ | $6.7 \pm 0.0$ | $4.35 \pm 0.24$ | $7.22 \pm 0.37$ | $4.57 \pm 0.24$ | $3.31 \pm 0.18$ | $3.30 \pm 0.23$ | $5.08 \pm 0.29$ |

Table 20: Dataset Size Ablation - Thinking Mode (ARC)

| Sample size | Accuracy | Thinking (%) | R Overall | R Accuracy | R Reasoning | R Comprehensive | R Pedagogic | R Confidence |
|---|---|---|---|---|---|---|---|---|
| 1000 | $0.7301 \pm 0.0261$ | $93.1 \pm 0.0$ | $6.75 \pm 0.15$ | $7.55 \pm 0.19$ | $7.03 \pm 0.14$ | $6.30 \pm 0.12$ | $6.51 \pm 0.14$ | $7.19 \pm 0.19$ |
| 2000 | $0.7543 \pm 0.0253$ | $94.8 \pm 0.0$ | $6.88 \pm 0.15$ | $7.71 \pm 0.18$ | $7.18 \pm 0.13$ | $6.46 \pm 0.10$ | $6.67 \pm 0.14$ | $7.38 \pm 0.18$ |

Table 21: Dataset Size Ablation - Standard Mode (ARC)

| Sample size | Accuracy | Thinking (%) | R Overall | R Accuracy | R Reasoning | R Comprehensive | R Pedagogic | R Confidence |
|---|---|---|---|---|---|---|---|---|
| 1000 | $0.6715 \pm 0.0326$ | $0.0 \pm 0.0$ | $6.64 \pm 0.20$ | $7.31 \pm 0.24$ | $6.97 \pm 0.18$ | $6.30 \pm 0.15$ | $6.54 \pm 0.19$ | $6.93 \pm 0.26$ |
| 2000 | $0.7633 \pm 0.0295$ | $0.0 \pm 0.0$ | $7.25 \pm 0.18$ | $7.97 \pm 0.22$ | $7.50 \pm 0.16$ | $6.73 \pm 0.14$ | $7.14 \pm 0.17$ | $7.73 \pm 0.23$ |

# C    Additional Related Work

Related work can be categorized into techniques for clarifying questions for multi-turn multimodal generation (MLLMs) or text-to-text generation (LLMs) settings. We also discuss related work on simulating user conversation trajectories and reinforcement learning approaches proposed for other problem settings.

## C.1    Supervised Learning

Many works have recently focused on clarifying user prompts by asking clarifying questions [39, 75]. Liu et al. [39] collect a dataset of 1,645 linguistic examples and different ambiguity labels. This is due to there being many different types of ambiguity. Zelikman et al. [75] introduced a simple and influential method: learn from rationales by fine-tuning on successful examples and regenerating rationales for failures. Given a prompt, generate a rationale and answer. If the answer is correct, fine-tune on prompt, rationale, and answer. Otherwise use the correct answer to generate a new rationale that leads to the correct answer. Fine-tune on prompt, rationale, and answer. This idea has since been extended in several directions. V-STaR [26] extends the idea to vision-language tasks, and Quiet-STaR [74] focuses on learning when not to ask, optimizing a policy to minimize unnecessary queries. We discuss extensions to reinforcement learning in Appendix C.4. A recent survey by Deng et al. [15] on proactive conversational techniques, which includes those focused on asking clarifying questions for disambiguation and the ilk.

Active disambiguation using LLMs has also been recently investigated [33, 79, 8]. AskToAct [79] focused on improving tool use via a self-correction mechanism for clarification. They generate a dataset and then fine-tune on it. Kobalczyk et al. [33] select clarifying questions based on information gain. Their approach emphasizes inference-time reasoning with pre-trained LLMs, while we learn task-specific policies that optimize questioning directly and efficiently without inference-time computation over all possible responses.

Recent works have also focused on benchmarking multi-turn conversational dialogue between users and agent for the purpose of clarification [8]. Zhang et al. [78] introduced a benchmark dataset and proposed an approach called Clarification-Execution-Planning (CEP) that uses specialized agents for clarification, execution, and planning. They predict if the question should be clarified and then generate a clarification.

Many works have also focused on the problem of predicting whether clarification is required in conversational interfaces [40, 8]. One recent work by [40] investigated a zero-shot approach for clarification detection in conversational search. They learn a classifier with an LLM backbone to predict if the query is specific or ambiguous. The training data are generated using a zero-shot LLM.

Li et al. [36] focuses on learning to ask critical questions in product search, using a dual-learning model that combines implicit session feedback with proactive clarification.

Surveys have further synthesized this area. Mulla and Gharpure [45] reviews progress in automatic question generation, including early reinforcement learning attempts, noting RL's ability to improve the flow of conversation by considering losses accumulated over $n$ turns in a dialog sequence. Furthermore, Zhang et al. [77] surveys how conversation analysis can help in the era of LLMs. They discussed conversation optimization using RL to improve conversation policy learning. The paper also touches on adapting LLMs with RL for goal-directed conversations, though not specifically focused on question asking.

## C.2  Supervised Learning with Multi-Modal Models

Multimodal multi-turn conversations that perform text-to-image generation have also been studied for asking clarifying questions to disambiguate and improve generation [20]. In particular, Hahn et al. [20] introduced an uncertainty-driven method that adaptively triggers clarifying questions when the system's confidence is low, enhancing multi-turn generation performance. This work also developed an automatic evaluation framework simulating users to assess question-asking strategies, using a suite of simple agents, including rule-based, belief-based, and LLM-based approaches, however, none of them incorporated any learning-based optimization.

Conversely, Villegas et al. [65] proposed ImageChain that focuses on image-to-text reasoning in MLLMs by considering a sequence of images as a multi-turn conversation along with the generated textual descriptions to create a succinct narrative, which has applications in video generation. Sequential reasoning over images and text. The description of the next image (treated as an agent) is conditioned on that image (treated as a user) and the history of the conversation.

Other work by Chen et al. [10] focused on improving multi-modal understanding for spoken conversations. They use spoken language to improve multi-modal conversations. That work constructed a dataset of per-turn preferences, annotating winning and losing responses, and applied Direct Preference Optimization (DPO) at each step. In contrast, our work improves upon this in three key ways: (1) we employ a more principled objective-driven simulation strategy; (2) we eliminate the need for DPO entirely since rewards are explicitly defined, direct reward-based policy gradients are both simpler and more efficient; and (3) we provide formal justification for our method.

## C.3  Classic RL

A large subset of prior work focuses on learning when and what to ask using RL. For example, DialFRED [19] trains an RL-based questioner agent to decide what questions to ask to complete household tasks, penalizing invalid questions. Sigaud et al. [59] used reinforcement learning to train an agent to ask questions. It uses question generation and question answering systems to create auxiliary objectives for reward shaping, improving sample efficiency in language-conditioned RL.

Further, Free et al. [17] leveraged Q-learning with DQN and BERT embeddings to train a chatbot that gathers hidden grid-world information by asking strategic questions to a simulated user. In the space of conversational recommendation, Lin [38] framed question selection as a bandit optimization problem, aiming to minimize unnecessary queries while also exploring RL fine-tuning of LLMs for human-like dialogue. Similarly, Wang and Ai [68] used reinforcement learning to train a DQN model for risk control in conversational search, focusing on when to ask clarifying questions. The RL agent learns to balance the rewards of asking relevant questions against the penalties of irrelevant ones.

Finally, Väth et al. [64] introduced a benchmark (LMRL-Gym) for evaluating multi-turn RL for LLMs, with the goal of enabling intentional interactions through carefully crafted questions, which is optimized by Q learning and DQN specifically.

## C.4  RL with LLMs

On RL with LLMs, Hong et al. [23] used offline RL to optimize goal-directed dialogues, leveraging LLMs to simulate human-like interactions and generate data for training. It addresses the limitations of LLMs in asking effective questions and optimizing for conversational outcomes over multiple turns. The method trains offline RL on the generated dataset. The RL algorithm is classic: implicit language q learning. We want to avoid value and Q functions.

One closely related work is learning to ask clarifying questions by STaR-GATE [2]. Their algorithm incorporates interactive conversations and preference elicitation from simulators, fine-tuning on best responses. This work leverages simulated trajectories between an optimized agent and a user to collect training data. Then it falls back to supervised learning: SFT on most rewarding trajectories is used to fine-tune the original LLM. This approach fails to make the full use of the reward signal, because SFT is equivalent to treating all best demonstrations as equally optimal. This leads to reduced statistical efficiency and a limited ability to capture nuanced training signals, which our approach addresses by preserving and exploiting the full reward structure.

Further, RL-STaR [9] provides a theoretical analysis for STaR-style updates in a reinforcement learning framework. Another related work is learning to tutor [55], which leverages simulated trajectories between an optimized agent and a user to collect training data. Then it applied DPO to learn from pairs of winning ans losing trajectories This approach fails to make the full use of the reward signal, since DPO reduces reward information to binary pairwise preferences, discarding finer-grained distinctions. This leads to reduced statistical efficiency and a limited ability to capture nuanced training signals, which our approach addresses by preserving and exploiting the full reward structure.

One work by Chen et al. [11] studied disambiguation in LLM-based conversations and develops an approach based on DPO for task-specific use cases that lack high-quality conversational trajectories such as data question-answering and SQL generation. Unlike the other works discussed above that focus on clarifying question generation for disambiguation in MLLMs, this work develops an approach for the simpler LLM clarification question generation problem that takes only text as input and generates only text as output (whether it is code, data, or other types of text). This is definitely RL. Similar to [55] but applied to multi-modal models. Additionally, Chi et al. [12] learned to ask clarifying questions in information retrieval. The key idea is to simulate potential clarifying questions and user responses, and then fine-tune on those that lead to the highest improvement in ranking metrics. This is not RL but the idea is similar to our SFT RL baseline.

Furthermore, Chu et al. [13] investigated SFT and RL on generalization and memorization and find that on a few text and visual tasks that RL generalizes better in both rule-based textual and visual environments whereas SFT mostly memorizes the training data and fails to generalize in the out-of-distribution setting. This one is methodological. Interestingly, we show a connection because RL can be viewed as weighted SFT. Another work by Arik et al. [3] improved conversational skills, specifically clarification question asking, using Action-Based Contrastive Self-Training (ACT). ACT is a DPO-based algorithm for sample-efficient dialogue policy learning. While RLHF is mentioned as a paradigm for building conversational agents, the paper's primary contribution is not directly about using RL for question asking, but DPO. Wang et al. [67] used reinforcement learning to enhance task-oriented dialogue systems, focusing on improving both understanding and generation tasks. It introduces step-by-step rewards throughout token generation to optimize dialogue state tracking and response generation. The approach is a variant of PPO and the focus is on individual token generation.

## C.5 Offline RL Algorithms for LLM Post-Training

It is well known in literature that when viewing an LLM based generation as a sequential decision process, the state comprises of the entire history of generated tokens, the action next generated token, and the transition function is a deterministic concatenation of the action token to the state tokens. So, when viewed from the perspective of an environment for RL, the only missing component is the reward function which is external to the LLM and needs to be provided. So, the key difference between online and offline RL in the case of LLMs is the availability of a reward function. In one of the earliest papers on RLHF [47], the authors converted offline feedback data collected from users to learn a reward function and then use an online RL algorithm (PPO) to train the LLM. Another branch of work attempted to explore use of offline RL methods to train LLMs with user feedback. One such method was ILQL [60], where the key idea was to learn a Q function with the LLM's hidden state forming the features for this Q function. In this case too some form of numerical reward from the user was needed, but this could be completely offline. The key considerations here were standard offline RL cautionary points such as ensuring to stay within the training data distribution for the Bellman updates (conservative QL) and the added complexity of estimating and using Q values during inference. Algorithms inspired by KL constrained policy optimization objectives such as DPO

[52] also function in an offline manner with the objective being to effectively learn an implicit reward function that is consistent with preference data collected from users. However, collection of pairwise preference data is a key requirement of this approach. A more detailed discussion on various offline policy based RL algorithms for LLM post-training is provided in Baheti et al. [4].

We specifically consider the objective functions of two policy based offline RL algorithms - DPO and ALOL to illustrate the key differences between them and our approach:

$$\nabla J(\theta)_{DPO} = \beta \mathbb{E}_{x \sim q, \tau_n \sim \pi_0(\cdot|x)}$$
$$\left[ \sigma(\hat{r}(x, \tau_{nl}) - \hat{r}(x, \tau_{nw})) \left[ \sum_{t=1}^{nw} \nabla \log \pi(a_t \mid x, \tau_{t-1}; \theta) - \sum_{t=1}^{nl} \nabla \log \pi(a_t \mid x, \tau_{t-1}; \theta) \right] \right],$$

$$\nabla J(\theta)_{A-LOL} = \mathbb{E}_{x \sim q, \tau_n \sim \pi_0(\cdot|x)} \left[ A_{\pi_0}(x, \tau_n) \hat{r}(x, \tau_n) \sum_{t=1}^{n} \nabla \log \pi(a_t \mid x, \tau_{t-1}; \theta) \right],$$

where $nw, nl$ represent the indices of the chosen and rejected sequences respectively, $\hat{r}$ represents the policy ratio of the propensities with respect to the reference policy and $A_{\pi_0}$ represents the advantage function under the reference policy. We notice that both these gradient estimates can be considered as scaled versions of the off-policy vanilla policy gradient, with the scaling factor in both these cases being a function of the ratio of the propensities under the policy being optimized and the reference policy. In our formulation, we avoid these scaling factors ensure stability and simplicity, while trading off for an objective that provides a loose lower bound for the original one.

## D  Dataset

In this section, we present a comprehensive summary of the six benchmark datasets discussed, along with the experimental setup:

**OpenBookQA [43]**  is a question-answering dataset modeled after open book exams, consisting of 5,957 multiple-choice elementary-level science questions (4,957 train, 500 dev, 500 test). It tests understanding of a small "book" of 1,326 core science facts and their application to novel situations. What makes this dataset challenging is that answering questions requires additional common knowledge beyond what's in the provided "book."

**SciQA [71]**  is a multimodal dataset that evaluates AI models' ability to reason using both textual and visual information for science topics. It includes approximately 21,000 multimodal questions covering physics, chemistry, and biology, sourced from educational materials. Models must analyze both text and diagrams to generate correct answers.

**MMLU [22]**  is a comprehensive benchmark that evaluates models on multiple choice questions across 57 subjects, including STEM, humanities, social sciences, and more, with difficulty levels ranging from elementary to advanced professional level. It focuses exclusively on zero-shot and few-shot settings, making it similar to how we evaluate humans. The benchmark tests both world knowledge and problem-solving ability.

**ARC [14]**  is a dataset of 7,787 genuine grade-school level, multiple-choice science questions from grade 3 to 9. It's divided into two parts: the Challenge Set with 2,590 "hard" questions that both retrieval and co-occurrence methods fail to answer correctly, and an Easy Set with 5,197 questions. Most questions have 4 answer choices, with less than 1% having 3 or 5 options. The dataset also includes a supporting knowledge base of 14.3 million unstructured text passages.

**CoSQL [73]**  is a corpus for building cross-domain conversational text-to-SQL systems. It consists of over 30,000 dialogue turns plus more than 10,000 annotated SQL queries, obtained from a Wizard-of-Oz collection of 3,000 dialogues querying 200 complex databases spanning 138 domains. Each dialogue simulates a real-world database query scenario with a crowd worker as a user exploring the database and a SQL expert retrieving answers with SQL. The average question length in CoSQL is 11.2 words with an average of 5.2 question turns per dialogue.

**MathDial [42]** is a dataset of one-to-one teacher-student tutoring dialogues grounded in multi-step math reasoning problems. The dataset contains 2,861 conversations in total, split into train and test sets. It was created by pairing human teachers with a Large Language Model (LLM) that was prompted to represent common student errors and uses LLMKT model [54]. The dataset focuses on effective tutoring rather than just problem-solving and exhibits rich pedagogical properties, focusing on guiding students using sense-making questions.

**Experimental Setup:** For our experiments, we randomly selected 500 samples from each dataset, allocating 400 for training and 100 for testing. We created conversations with 3 turns and generated 3 random runs (trajectories) with different temperature settings using our `Base` model.

# E   Multi-Turn Reasoning Prompts and Conversations

Our multi-turn reasoning experiments encourage the assistant to progressively deepen its analysis through iterative prompting by the teacher. The default conversation length is 3 turns, although we also experiment with longer conversations:

1. **Turn 1 - Initial Question:** Teacher presents the problem and asks for an initial analysis.
2. **Turn 1 - Initial Response:** Assistant replies with initial thoughts.
3. **Turn 2 - Deeper Analysis:** Teacher prompts for wrong options and key concepts.
4. **Turn 2 - Deeper Analysis Response:** Assistant replies with a detailed reasoning.
5. **Turn 3 - Final Answer:** Teacher asks for the final answer with full justification.
6. **Turn 3 - Final Response:** Assistant replies with the final answer and its justification.

Next we present detailed prompts for these conversations, first **with thinking tags** and then **without thinking tags**.

## E.1   Prompts WITH Thinking Tags

---

**System Prompt - Assistant (With Thinking)**

*You are a helpful, accurate assistant who is an expert at answering multiple-choice questions. When you think through a problem, wrap your thinking in <thinking></thinking> tags. You MUST first state your final answer in the format: 'The answer is X' where X is A, B, C, or D. The final answer must be outside the thinking tags. Then show your thinking in <thinking></thinking> tags for your step-by-step reasoning.*

---

**Turn 1 - Teacher Initial Question (With Thinking)**

```
Question:  [question text]
Choices:
A. [choice A]
B. [choice B]
C. [choice C]
D. [choice D]

[Dataset-specific instruction]
```

*Please use <thinking></thinking> tags to show your step-by-step reasoning, then provide your initial thoughts outside of these tags.*

---

**Turn 1 - Assistant Initial Response (With Thinking)**

Initial thoughts or analysis (outside thinking tags) `<thinking>Step-by-step reasoning about each option</thinking>`

---

**Turn 2 - Teacher Deeper Analysis (With Thinking)**

*That's a good start. Can you explain more about why some options might be incorrect? Use <thinking></thinking> tags for your analysis.*

**Turn 2 - Assistant Deeper Analysis Response (With Thinking)**

Detailed elimination reasoning (outside thinking tags) `<thinking>Systematic analysis of why wrong options fail and why correct option succeeds</thinking>`

**Turn 3 - Teacher Final Answer (With Thinking)**

*Thank you for your detailed explanations. What is your final answer (A, B, C, or D)? Please provide a justification for your choice. You MUST first state your final answer in the format: 'The answer is X' where X is A, B, C, or D. The final answer must be outside the thinking tags. Then show your thinking in <thinking></thinking> tags for your step-by-step reasoning.*

**Turn 3 - Assistant Final Response (With Thinking)**

`The answer is X.          <thinking>Complete justification with step-by-step reasoning</thinking>`

For longer conversations than 3 turns, we use the following intermediate prompt between the second and final turns.

**Intermediate Turn - Teacher (With Thinking)**

*Thank you for your explanation. Let's explore this further (turn [N]). Could you elaborate on the key concepts relevant to this question? Continue using <thinking></thinking> tags for your analysis.*

## E.2 Prompts WITHOUT Thinking Tags

**System Prompt - Assistant (Without Thinking)**

*You are a helpful, accurate assistant who is an expert at answering multiple-choice questions. You MUST first state your final answer in the format: 'The answer is X' where X is A, B, C, or D. After the final answer clearly explain your reasoning.*

**Turn 1 - Teacher Initial Question (Without Thinking)**

```
Question:  [question text]
Choices:
A. [choice A]
B. [choice B]
C. [choice C]
D. [choice D]

[Dataset-specific instruction]
```

*Please think through this step by step and explain your initial thoughts about the question.*

> **Turn 1 - Assistant Initial Response (Without Thinking)**
>
> Step-by-step reasoning and initial thoughts (all outside, no thinking tags)

> **Turn 2 - Teacher Deeper Analysis (Without Thinking)**
>
> *That's a good start. Can you explain more about why some options might be incorrect? Also, are there any key concepts or facts that are relevant to answering this question?*

> **Turn 2 - Assistant Deeper Analysis Response (Without Thinking)**
>
> Detailed elimination reasoning with explanations (all outside, no thinking tags)

> **Turn 3 - Teacher Final Answer (Without Thinking)**
>
> *Thank you for your detailed explanations. What is your final answer (A, B, C, or D)? You MUST first state your final answer in the format: 'The answer is X' where X is A, B, C, or D. After the final answer clearly explain your reasoning.*

> **Turn 3 - Assistant Final Response (Without Thinking)**
>
> ```
> The answer is X.
> ```
> Complete justification with step-by-step reasoning (all outside, no thinking tags)

For longer conversations than 3 turns, we use the following intermediate prompt between the second and final turns.

> **Intermediate Turn - Teacher (With Thinking)**
>
> *Thank you for your explanation. Let's explore this further (turn [N]). Could you elaborate on the key concepts relevant to this question?*

## E.3   Evaluation Framework

All evaluation is done by a **GPT-4o judge**. The judge is provided with the complete conversation transcript up to the last turn, along with the initial question, choices, and correct answer.

We consider the following 6 evaluation metrics:

1. **Accuracy** (0-10) - Is the assistant moving toward the correct answer?

2. **Reasoning ability** (0-10) - Is the reasoning logical, clear, and precise?

3. **Comprehensiveness** (0-10) - Are key concepts being properly explained?

4. **Pedagogical value** (0-10) - Is this explanation helping someone learn?

5. **Confidence calibration** (0-10) - Is the assistant appropriately confident?

6. **Overall** (0-10) - Holistic assessment of conversation quality.

## E.4  ARC Conversation with `Base`

We first present a 3-turn conversation with `Base` on the ARC dataset. The final response shows that `Base` is very verbose.

**Teacher Message 1**

Question: Which of the following best explains how stems transport water to other parts of the plant? Choices: A. through a chemical called chlorophyll B. by using photosynthesis C. through a system of tubes D. by converting water to food
This question is from a science exam. Please use <thinking></thinking> tags to show your step-by-step reasoning, then provide your initial thoughts outside of these tags.

**Assistant Message 1**

The answer is C. <thinking> To determine the correct answer, I will evaluate each option in relation to how stems transport water to other parts of a plant. A. Chlorophyll is a chemical that is essential for photosynthesis, but it is not directly responsible for transporting water through the plant. Chlorophyll is involved in absorbing light energy, which is necessary for photosynthesis to occur, but it is not the mechanism by which water is transported. B. Photosynthesis is the process by

**Teacher Message 2**

That's a good start. Can you explain more about why some options might be incorrect? Use <thinking></thinking> tags for your analysis.

**Assistant Message 2**

<thinking> Photosynthesis is the process by which plants convert light energy into chemical energy in the form of glucose. While photosynthesis occurs in the leaves, which are attached to the stems, it is not the process by which water is transported to other parts of the plant. The question specifically asks about the mechanism of water transport, not the process of energy conversion. C. The xylem is a type of vascular tissue that is found in the stems of plants, which is responsible for transporting water and

**Teacher Message 3**

Thank you for your detailed explanations. What is your final answer (A, B, C, or D)? Please provide a justification for your choice. You MUST first state your final answer in the format: 'The answer is X' where X is A, B, C, or D. The final answer must be outside the thinking tags. Then show your thinking in <thinking></thinking> tags for your step-by-step reasoning.

**Assistant Message 3**

</thinking> The xylem consists of dead cells that are hollow, which allows water to pass through them. This is why option A is incorrect. Chlorophyll is a chemical used in photosynthesis, not in transporting water. Photosynthesis is the process of plants converting light energy into chemical energy, but it's not related to water transport. Therefore, option B is also incorrect. Option D is incorrect because converting water to food is the process of photosynthesis

**Evaluation Scores by GPT4o**

| Criteria | Score | Rating |
|---|---|---|
| Accuracy | 8/10 | |
| Reasoning Ability | 7/10 | |
| Comprehensiveness | 6/10 | |
| Pedagogical Value | 7/10 | |
| Confidence Calibration | 6/10 | |
| **Overall** | **7.0/10** | |

**Evaluation Justifications by GPT4o**

**Accuracy: 8/10**

The assistant is moving toward the correct answer, "C," and has correctly identified that the xylem is responsible for water transport. However, the explanation for option D is incomplete, and the assistant has not yet explicitly stated the final answer as requested.

**Reasoning Ability: 7/10**

The reasoning is mostly logical and clear, particularly in explaining why options A and B are incorrect. However, the analysis of option D is cut off and incomplete, which detracts from the overall clarity and precision of the reasoning.

**Comprehensiveness: 6/10**

The assistant provides a good explanation for why options A and B are incorrect and begins to explain why C is correct. However, the discussion of option D is incomplete, and the explanation of the xylem could be expanded further to fully address the mechanism of water transport.

**Pedagogical Value: 7/10**

The conversation is educational and provides some useful insights, particularly about chlorophyll and photosynthesis. However, the incomplete explanations for D and the xylem system limit the overall learning potential.

**Confidence Calibration: 6/10**

The assistant appears confident in its reasoning but has not yet explicitly stated the final answer as requested. Additionally, the incomplete explanation of option D suggests a slight overconfidence in the clarity of its analysis.

**Overall Assessment by GPT4o**

# Final Score: 7.0/10

**Summary:** The conversation is on the right track and provides a mostly accurate and logical explanation of the question. However, incomplete reasoning for option D and a lack of explicit final answer detract from the overall quality. Expanding on the xylem's role and fully addressing all options would improve the conversation significantly.

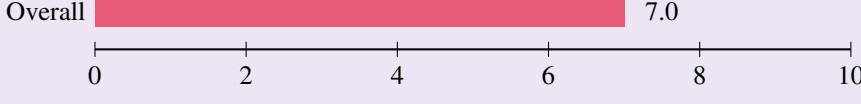

## E.5 ARC Conversation with `Swift`

Now we show the conversation of `Swift` on the same ARC dataset. The teacher messages are the same and hence we do not show them. We observe that `Swift` updated its reasoning process with additional information and thus gets a higher evaluation score from the GPT-4o judge.

**Assistant Message 1**

«SYS» The answer is C.<thinking>First, I considered each choice individually. I looked at A and thought, 'What does chlorophyll do?' Chlorophyll is the green pigment in plants that helps them make food from sunlight through photosynthesis, so this choice doesn't sound right. Next, I looked at B and thought, 'What is photosynthesis?' Photosynthesis is the process plants use to make food from sunlight, so this choice also doesn't seem correct. Then

**Assistant Message 2**

<thinking>I examined C and thought, 'Is there a system of tubes in a plant that could transport water?' Yes, I know that plants have a system of tubes called xylem that transport water from the roots to the rest of the plant. Finally, I looked at D and thought, 'Does water get converted to food in the process of transporting it?' No, that's not how it works. Plants use water to make food, but that's not what's happening when water is being transported

**Assistant Message 3**

choice C is the best explanation.
</thinking>I have now fully explained why choice C is the best explanation for how stems transport water to other parts of the plant. Choice C states that stems transport water through a system of tubes, which is a more accurate description of the plant's water transport system. This system, known as xylem, is a network of tubes that allows water to move from the roots to the leaves of the plant. This is a fundamental process that supports the plant's

**Evaluation Scores by GPT4o**

| Criteria | Score | Rating |
|---|---|---|
| Accuracy | 10/10 | |
| Reasoning Ability | 10/10 | |
| Comprehensiveness | 10/10 | |
| Pedagogical Value | 10/10 | |
| Confidence Calibration | 10/10 | |
| **Overall** | **10.0/10** | |

## E.6 CoSQL Conversation with `Base`

Our next example is a 3-turn conversation with `Base` on the CoSQL dataset. The final response shows that `Base` follows reasoning from the previous turns. It moves towards the correct answer but the reasoning is not comprehensive.

**Teacher Message 1**

Question: Your task is to generate SQL query based on the following question and information provided. Which distinctive models are produced by maker with the full name General Motors or weighing more than 3500? Hint: Consider relationships between multiple tables. The relevant schema tables are: ['car_makers', 'model_list', 'car_names', 'cars_data']. The optional foreign keys are: ['CAR_NAMES.model = MODEL_LIST.model', 'MODEL_LIST.maker = CAR_MAKERS.id', 'CARS_DATA.id = CAR_NAMES.makeid']. Choices: A. A. SELECT DISTINCT T1.Model, T3.FullName FROM CARS_DATA AS T4 JOIN CAR_NAMES AS T1 ON T4.Id = T1.MakeId LEFT JOIN CAR_MAKERS AS T3 ON T1.MakeId = T3.Id WHERE T3.FullName LIKE '%General%' OR T4.weight < 3500; B. B. SELECT DISTINCT Model FROM MODEL_LIST WHERE Maker IN (SELECT Id FROM CAR_MAKERS WHERE FullName = 'General Motors' AND Id IN (SELECT MakeId FROM CARS_DATA WHERE weight > 3500)); C. C. SELECT DISTINCT T2.Model FROM CAR_NAMES AS T1 JOIN MODEL_LIST AS T2 ON T1.Model = T2.Model JOIN CAR_MAKERS AS T3 ON T2.Maker = T3.Id JOIN CARS_DATA AS T4 ON T1.MakeId = T4.Id WHERE T3.FullName = 'General Motors' OR T4.weight > 3500; D. D. SELECT T1.Model FROM CAR_NAMES AS T1 WHERE T1.Model IN (SELECT Model FROM CARS_DATA WHERE weight > 3500) AND T1.MakeId IN (SELECT Id FROM CAR_MAKERS WHERE FullName = 'General Motors');
This question asks for the correct SQL query to retrieve the requested information from a database. Focus on SQL syntax, correct table joins, and query efficiency. Please think through this step by step and explain your initial thoughts about the question.

**Assistant Message 1**

«SYS» The answer is C.
To determine the correct SQL query, let's break down the requirements step by step:
1. **Distinctive models**: We need to find distinct models that match the given conditions. This means we should use the 'SELECT DISTINCT' clause.
2. **Maker with full name General Motors**: This indicates we need to filter the data based on the 'FullName' of the maker, which is stored in the 'CAR_MAKERS' table.
3.

**Teacher Message 2**

That's a good start. Can you explain more about why some options might be incorrect? Also, are there any key concepts or facts that are relevant to answering this question?

**Assistant Message 2**

«SYS»
To evaluate the options, let's consider the following key concepts and facts:
* **Joining tables**: We need to join multiple tables to retrieve the required information. The correct join order and type are crucial. * **Filtering conditions**: We have two conditions to apply: the maker's full name should be 'General Motors' or the car's weight should be more than 3500. * **Using foreign keys**: The foreign keys are defined as 'CAR

**Teacher Message 3**

Thank you for your detailed explanations. What is your final answer (A, B, C, or D)? You MUST first state your final answer in the format: 'The answer is X' where X is A, B, C, or D. After the final answer clearly explain your reasoning.

## Assistant Message 3

<SYS> group by clause, so option D is not correct. Now, let's consider the joins and subqueries in the other options. The answer is C. To solve this problem, we need to find the instructors who did not teach any courses. This can be achieved by identifying instructors who do not have a matching id in the 'teaches' table.

We can use a subquery to select the ids of instructors who taught courses. Then, we can use the

## Evaluation Scores by GPT4o

| Criteria | Score | Rating |
|----------|-------|--------|
| Accuracy | 7/10 | |
| Reasoning Ability | 6/10 | |
| Comprehensiveness | 5/10 | |
| Pedagogical Value | 6/10 | |
| Confidence Calibration | 7/10 | |
| **Overall** | **6.0/10** | |

## Evaluation Justifications by GPT4o

**Accuracy: 7/10**

The assistant is moving toward the correct answer (C) and has identified it as the correct choice. However, the explanation provided so far is incomplete, and the assistant has not yet fully justified why C is correct or why the other options are incorrect.

**Reasoning Ability: 6/10**

The reasoning is partially logical and clear, as the assistant has identified the need for 'SELECT DISTINCT', proper joins, and filtering conditions. However, the explanation lacks depth and precision, particularly in explaining the relationships between tables and why certain options fail to meet the requirements.

**Comprehensiveness: 5/10**

Key concepts like table joins, filtering conditions, and foreign key relationships are mentioned, but they are not fully explained. The assistant has not yet addressed why specific options (A, B, and D) are incorrect, which is critical for a comprehensive analysis.

**Pedagogical Value: 6/10**

The explanation has some educational value, as it introduces important SQL concepts like 'SELECT DISTINCT', filtering, and table joins. However, the lack of detailed reasoning and comparison between options limits its effectiveness as a learning resource.

**Confidence Calibration: 7/10**

The assistant confidently identifies C as the correct answer, which is accurate. However, the confidence is slightly undermined by the incomplete reasoning and lack of detailed analysis of the other options.

**Overall Assessment by GPT4o**

# Final Score: 6.0/10

**Summary:** The assistant is on the right track and has correctly identified the answer, but the explanation lacks depth, clarity, and comprehensiveness. While some key concepts are mentioned, the conversation would benefit from a more thorough analysis of why the other options are incorrect and a clearer breakdown of the relationships between tables.

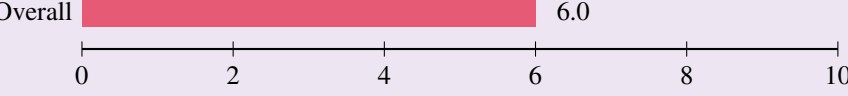

## E.7 CoSQL Conversation with `Swift`

Now we show the conversation of `Swift` on the same CoSQL dataset. The teacher messages are the same and hence we do not show them. We observe that the response is more accurate than that of `Base`, as judged by GPT-4o.

---

**Assistant Message 1**

«SYS»[/INST]The answer is C.<thinking>To answer this question, we need to follow these steps:1. Identify the tables and fields involved.2. Determine the relationship between the tables. 3. Choose the correct join type.4. Specify the conditions for the query.5. Order the results to find the model with the largest horsepower.1. The relevant tables are 'car_names' and 'cars_data'.

---

**Assistant Message 2**

«SYS»[/INST]<thinking>The correct query should involve the 'car_names' table for the model information and the 'cars_data' table for the horsepower information. The relationship between the tables is established by the'makeid' field, which is the foreign key in 'car_names' and the primary key in 'cars_data'.Looking at the options, we can eliminate those that don't use the correct join type or don't correctly filter the results.A.

---

**Assistant Message 3**

«SYS» The answer is C.
To determine the correct SQL query, let's break down the requirements step by step:
1. **Distinctive models**: We need to find distinct models that match the given conditions. This means we should use the `SELECT DISTINCT` clause.
2. **Maker with full name General Motors**: This indicates we need to filter the data based on the `FullName` of the maker, which is stored in the `CAR_MAKERS` table.
3.

---

**Evaluation Scores by GPT4o**

| Criteria | Score | Rating |
|---|---|---|
| Accuracy | 8/10 | |
| Reasoning Ability | 7/10 | |
| Comprehensiveness | 6/10 | |
| Pedagogical Value | 5/10 | |
| Confidence Calibration | 7/10 | |
| **Overall** | **6.6/10** | |

**Evaluation Justifications by GPT4o**

**Accuracy: 8/10**

The response correctly identifies the correct answer as "C" and provides reasoning aligned with the question's requirements. However, the explanation is incomplete and does not fully justify why "C" is correct or why other options are incorrect.

**Reasoning Ability: 7/10**

The reasoning is partially sound, as it breaks down the requirements of the query and links them to the SQL components. However, the explanation is truncated and does not fully analyze the relationships between tables or the logic of the query.

**Comprehensiveness: 6/10**

The response addresses some key aspects of the question, such as the need for 'SELECT DISTINCT' and filtering by 'FullName'. However, it does not explore the schema relationships, optional foreign keys, or why the other options fail to meet the requirements.

**Pedagogical Value: 5/10**

The response provides some educational value by explaining the use of 'SELECT DISTINCT' and filtering conditions. However, it lacks depth and does not guide the reader through the full reasoning process or clarify why the correct query works.

**Confidence Calibration: 7/10**

The response confidently states that the correct answer is "C" and provides some justification. However, the incomplete explanation leaves room for doubt about whether the reasoning is fully understood.

---

**Overall Assessment by GPT4o**

# Final Score: 6.6/10

**Summary:** While the response identifies the correct answer and provides some reasoning, it lacks thoroughness, depth, and a complete analysis of the question and options. Improvements in comprehensiveness and pedagogical clarity are needed.

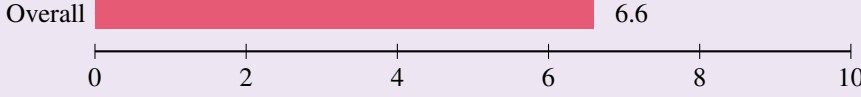

---

### E.8 MathDial Conversation with `Base`

Our final example is a 3-turn conversation with `Base` on the MathDial dataset.

---

**Teacher Message 1**

Drew is reseeding his lawn with grass seed. One bag of grass seed covers 250 square feet of lawn. His lawn is 22 feet from the house to the curb and 36 feet from side to side. He bought four bags of seed. How many extra square feet could the leftover grass seed cover after Drew reseeds his lawn?

A student attempted to solve this problem as follows: The area of Drew's lawn is 22 x 36 = 792 square feet. Four bags of grass seed can cover 4 x 250 = 1000 square feet. Drew will use all the grass seed to cover his lawn, so there won't be any leftover to cover extra square feet. Therefore, the leftover grass seed can't cover any extra square feet. 0

Which of the following is the correct final answer? A. 177 B. 0 C. 138 D. 208

Please use <thinking></thinking> tags to show your step-by-step reasoning.

## Assistant Message 1

«SYS»[/SYS]
The answer is B. <thinking> The student correctly calculated the area of the lawn, which is $22 \times 36 = 792$ square feet. Then, they calculated the total area that the four bags of grass seed could cover, which is $4 \times 250 = 1000$ square feet. However, they incorrectly concluded that there would be no leftover grass seed because the area of the lawn (792 square feet) is less than the total area that the grass seed could cover (1000 square feet). This

## Teacher Message 2

Can you explain more about why some options might be incorrect? Use <thinking></thinking> tags for your analysis.

## Assistant Message 2

«/SYS»
The answer is B. <thinking> The student's conclusion is incorrect because they didn't take into account the fact that the lawn is a rectangle and the grass seed can only cover the area of the lawn. The total area that the grass seed can cover (1000 square feet) is greater than the area of the lawn (792 square feet), but that doesn't mean that the entire lawn can be covered with the grass seed. In fact, the student's calculation

## Teacher Message 3

Thank you for your detailed explanations. What is your final answer (A, B, C, or D)? Please provide a justification for your choice. You MUST first state your final answer in the format: 'The answer is X' where X is A, B, C, or D. The final answer must be outside the thinking tags. Then show your thinking in <thinking></thinking> tags for your step-by-step reasoning.

## Assistant Message 3

«/SYS»
The answer is B.
<thinking> The student's conclusion is incorrect because they didn't take into account the fact that the lawn is a rectangle and the grass seed can only cover the area of the lawn. The total area that the grass seed can cover (1000 square feet) is

## Evaluation Scores by GPT4o

| Criteria | Score | Rating |
|---|---|---|
| Accuracy | 3/10 | |
| Reasoning Ability | 4/10 | |
| Comprehensiveness | 3/10 | |
| Pedagogical Value | 4/10 | |
| Confidence Calibration | 2/10 | |
| **Overall** | **3.0/10** | |

## E.9  MathDial Conversation with `Swift`

Now we show the conversation of `Swift` on the same MathDial dataset. The teacher messages are the same and hence we do not show them. We observe that the response is more accurate than that of `Base`, as judged by GPT-4o.

**Assistant Message 1**

«SYS»<thinking>Let's analyze the options to see why they might be incorrect.Option B (23 hours) is incorrect because it doesn't take into account the fact that Mike watches TV for 4 hours every day. If he only played video games for 2 hours every day, the total time spent playing video games would be 2 x 3 = 6 hours, and the total time spent watching TV would be 4 x 7 = 28 hours"

### Assistant Message 2

<SYS>[/SYS]<thinking>Let's analyze the incorrect options.Option A: 40 hours. This option is incorrect because it is more than the total number of hours Mike spends watching TV and playing video games. We calculated earlier that Mike spends 46 hours watching TV and playing video games.Option B: 23 hoursThis option is incorrect because it is less than the total number of hours Mike spends watching TV and playing video games.

### Assistant Message 3

«SYS»[/SYS] <thinking>The student's incorrect conclusion that there would be no leftover grass seed is based on the fact that the area of the lawn is less than the total area that the grass seed could cover. However, this does not necessarily mean that there will be no leftover grass seed. In fact, there will be 1000 - 792 = 208 square feet of leftover grass seed. This is because the four bags of grass seed can cover 1000 square

### Evaluation Scores by GPT4o

| Criteria | Score | Rating |
|---|---|---|
| Accuracy | 5/10 | |
| Reasoning Ability | 6/10 | |
| Comprehensiveness | 4/10 | |
| Pedagogical Value | 5/10 | |
| Confidence Calibration | 3/10 | |
| **Overall** | **5.0/10** | |

### Evaluation Justifications by GPT4o

**Accuracy: 5/10**

The assistant has correctly calculated the area of the lawn (792 square feet) and the total coverage of four bags of grass seed (1000 square feet). However, the assistant prematurely states that the answer is "A" without fully completing the calculations or verifying the leftover area, which leads to an incorrect conclusion.

**Reasoning Ability: 6/10**

The assistant demonstrates some logical reasoning by breaking the problem into steps (calculating the lawn area and total coverage). However, the reasoning is incomplete, as the assistant does not finish analyzing the leftover area or fully explain why "A" is chosen.

**Comprehensiveness: 4/10**

The explanation is missing key steps, such as subtracting the lawn area from the total coverage to determine the leftover area (1000 - 792 = 208). Additionally, the assistant does not fully analyze all the options, leaving the explanation incomplete and confusing.

**Pedagogical Value: 5/10**

While the assistant starts with a structured approach, the incomplete reasoning and incorrect conclusion could mislead a learner. The explanation lacks clarity and depth, which limits its educational value.

**Confidence Calibration: 3/10**

The assistant confidently states that the answer is "A" without completing the necessary calculations or fully analyzing the problem. This overconfidence is unwarranted given the incomplete reasoning.

## F    Multi-Turn Clarifying Questions Prompts and Conversations

Our clarifying questions experiments are 3-turn conversations between an assistant and teacher:

1. Teacher presents presents the question (hard-coded).
2. Assistant asks a clarifying question about one option.
3. Teacher responds to the question.
4. Assistant thinks step by step.
5. Teacher asks for the final answer with full justification (hard-coded).
6. Assistant replies with the final answer and its justification.

Next we present detailed prompts for these conversations. We use the same GPT-4o judge as in Appendix E.

### F.1    Prompts

---

**System Prompt - Teacher**

*You are an expert teacher helping a student learn through clarifying questions. Guide the student by presenting questions clearly, answering their clarifications helpfully, and encouraging deeper thinking. NEVER give away answers - let the student discover through questioning. Keep your responses concise and focused.*

---

**System Prompt - Assistant**

*You are a curious student learning through asking clarifying questions. Think carefully about what you need to understand, ask specific questions to clarify concepts, and use the teacher's guidance to arrive at well-reasoned answers. For thinking steps, use lowercase <thinking>your thoughts</thinking> tags exactly. For final answers, format as: Answer is X. <thinking>your reasoning</thinking> (lowercase thinking tags only).*

---

**Turn 1 - Teacher Initial Question (Hard-Coded)**

```
Question: [question text]
A) [choice A]
B) [choice B]
C) [choice C]
D) [choice D]

Ask me ONE specific question about any ONE of the options
(A, B, C, or D). What would you like to clarify?
```

---

## F.2 Comparison: Multi-Turn Reasoning Versus Clarifying Questions

We summarize differences between our reasoning and clarifying questions settings in Table 22.

## F.3 MMLU Conversation with `Base`

We first show a 3-turn conversation with `Base` on the MMLU dataset.

> **Teacher Message 1**
>
> Question: The "quantum jump" mentioned in the passage most directly contributed to which of the following developments in the period 1450–1750 C.E.? A) A breakdown in trade routes through the collapse of the established state structure B) An increase in the population of the world through more plentiful supplies of food C) The spread of Chinese and Indian belief systems across the world D) An increase in social unrest

> **Assistant Message 1**
>
> I'd like to clarify: Does option A refer specifically to the collapse of trade routes due to European colonialism or does it imply a broader collapse of the system.

| Aspect | Multi-Turn Reasoning | Multi-Turn Clarification |
|---|---|---|
| **Conversation Type** | User-Assistant dialogue | Teacher-Assistant dialogue |
| **Number of Turns** | 3+ (configurable) | Fixed 6 steps |
| **Interaction Style** | User prompts progressively deeper reasoning | Assistant asks questions, teacher guides |
| **Turn 1** | User presents question | Teacher presents question |
| **Turn 2** | User: "Explain why options are incorrect" | Assistant asks clarification question |
| **Turn 3** | User: "What's your final answer?" | Teacher answers clarification |
| **Turn 4+** | [Optional: more exploration] / Final answer | Assistant thinks step-by-step |
| **Initiative** | User drives (external prompting) | Assistant drives (internal questioning) |
| **Learning Style** | Guided reflection | Active inquiry |
| **Models Involved** | Single model (assistant) | Two roles (Assistant to be trained, base teacher) |
| **Training Goal** | Learn to reason deeply through iterative prompting | Learn to ask good clarifying questions |
| **Evaluation** | Same 6 metrics, same GPT-4o evaluator | Same 6 metrics, same GPT-4o evaluator |
| **Answer Format** | "The answer is X" + thinking | "Answer is X" + thinking |
| **Primary Use Case** | Direct reasoning improvement | Interactive learning through questioning |

Table 22: Comparison between Multi-Turn Reasoning and Clarification approaches

---

**Teacher Message 2**

The passage actually refers to a more general concept, implying a larger-scale disruption in global economic connections rather than just focusing on European colonial expansion. Would you like me to explain further?

---

**Assistant Message 2**

<thinking>option A seems plausible since it relates to disruptions caused by colonial expansion. However, the passage doesn't explicitly mention trade route collapse, so we should consider other options more closely. Option B appears unlikely given the context - population growth doesn't necessarily follow from exploitation and technological advancements alone.</thinking> <thinking>Option C seems unlikely as well because there's no indication that belief systems spread due

---

**Teacher Message 3**

Great. Based on the discussion so far think and reason more and give your final answer. You must give the answer first then explain your reasoning within thinking tags.

---

**Assistant Message 3**

Answer is A <thinking>thinking>i believe option A is correct because the passage suggests that the exploitation of resources led to disruptions in existing trade networks, which fits with the idea of a breakdown in trade routes. This makes sense given the context of European colonization and its impact on global economic connections.</thinking>

| Criteria | Score | Rating |
|----------|-------|--------|
| Accuracy | 4/10 | ■ ■ ■ ■ ◻ ◻ ◻ ◻ ◻ ◻ |
| Reasoning Ability | 5/10 | ■ ■ ■ ■ ■ ◻ ◻ ◻ ◻ ◻ |
| Comprehensiveness | 4/10 | ■ ■ ■ ■ ◻ ◻ ◻ ◻ ◻ ◻ |
| Pedagogical Value | 5/10 | ■ ■ ■ ■ ■ ◻ ◻ ◻ ◻ ◻ |
| Confidence Calibration | 3/10 | ■ ■ ■ ◻ ◻ ◻ ◻ ◻ ◻ ◻ |
| **Overall** | **4.2/10** | ■ ■ ■ ■ ◻ ◻ ◻ ◻ ◻ ◻ |

**Evaluation Justifications -** `Base`

**Accuracy: 4/10**

Wrong answer (A). Student misinterpreted the passage's focus. The passage discusses exploitation leading to INCREASED productivity and resources, not breakdown of trade. Student incorrectly focused on "disruptions" rather than positive outcomes.

**Reasoning Ability: 5/10**

Flawed reasoning. Student acknowledged "passage doesn't explicitly mention trade route collapse" but still chose option A. Incorrectly dismissed option B ("population growth doesn't necessarily follow") despite strong textual support.

**Comprehensiveness: 4/10**

Failed to connect key passage elements: exploitation of ecosystems → agricultural productivity → food supplies → population. Focused too narrowly on "disruptions" without considering constructive outcomes.

**Pedagogical Value: 5/10**

Limited educational value. Demonstrated how to consider options but arrived at wrong conclusion by misinterpreting passage's emphasis. Could mislead learners about historical causation.

**Confidence Calibration: 3/10**

Poor confidence calibration. Expressed high certainty ("I believe option A is correct") despite acknowledging lack of textual support. Should have been more uncertain given weak evidence.

**Overall Assessment by GPT4o**

# Final Score: 4.2/10

**Summary:** The assistant misinterpreted the passage's focus on productivity increase, incorrectly selecting trade route collapse instead of population growth. Poor confidence calibration was evident with high certainty expressed despite acknowledging lack of textual support, and the failure to connect exploitation to agricultural improvements significantly undermined the response quality.

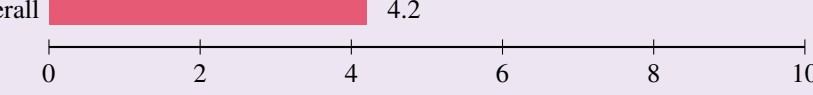

## F.4 MMLU Conversation with `Refit`

Next we show a 3-turn conversation with `Refit` on the MMLU dataset.

**Teacher Message 1**

Question: The "quantum jump" mentioned in the passage most directly contributed to which of the following developments in the period 1450–1750 C.E.? A) A breakdown in trade routes through the collapse of the established state structure B) An increase in the population of the world through more plentiful supplies of food C) The spread of Chinese and Indian belief systems across the world D) An increase in social unrest

**Assistant Message 1**

What does "quantum leap" mean in relation to historical events, particularly when referring to the development of new technologies?

**Teacher Message 2**

[ INST ] You asked a great clarifying question! You like to know more about what I meant by "a quantum leap"? ## Step 1: Understand the context The term "quantum leap" is used here which means there has been significant progress in this area since its inception.

**Assistant Message 2**

<Thinking>The first thing to consider when evaluating A is whether the collapse of established state structures could be caused by an increase in productivity due to exploiting natural resources and human labor outside of Europe.</Thinking> <Thinking>The idea that increased productivity leads to population growth seems plausible because with higher yields and better technology, people can produce more food, leading to an uptick in population.</Thinking> <Thinking>This

**Teacher Message 3**

Great. Based on the discussion so far think and reason more and give your final answer. You must give the answer first then explain your reasoning within thinking tags.

**Assistant Message 3**

Answer is B. <thinking>Thinking this makes sense because the text mentions "exploiting the ecosystems" which suggests using natural resources to improve agricultural output; thus leading to better food production and increased population growth. The other options don't fit as strongly based on our previous discussion.</thinking>

**Evaluation Scores - Refit**

| Criteria | Score | Rating |
|---|---|---|
| Accuracy | 8/10 | |
| Reasoning Ability | 7/10 | |
| Comprehensiveness | 6/10 | |
| Pedagogical Value | 7/10 | |
| Confidence Calibration | 6/10 | |
| **Overall** | **6.8/10** | |

**Evaluation Justifications -** `Refit`

**Accuracy: 8/10**

Correctly identified answer B. Student made the crucial connection between "exploiting ecosystems" and improved agricultural output leading to population growth through increased food supplies.

**Reasoning Ability: 7/10**

Strong logical reasoning. Student systematically considered multiple options, correctly eliminated option A, and identified the causal chain: exploitation → better agriculture → more food → population growth.

**Comprehensiveness: 6/10**

Covered key concepts connecting productivity increase, agricultural exploitation, and demographic changes. Could have been more explicit about eliminating other options.

**Pedagogical Value: 7/10**

Good educational value. The reasoning process demonstrated how to connect historical evidence (passage) with potential outcomes (answer choices) through logical analysis.

**Confidence Calibration: 6/10**

Appropriate confidence. Statement "makes sense" shows reasonable certainty backed by textual evidence, while acknowledging consideration of other options.

---

**Overall Assessment by GPT4o**

# Final Score: 6.8/10

**Summary:** The assistant showed strong logical reasoning by correctly identifying the causal chain from exploitation to agricultural output to population growth. The systematic consideration of multiple options and connection between textual evidence and answer choice demonstrated solid analytical skills, resulting in the correct answer.

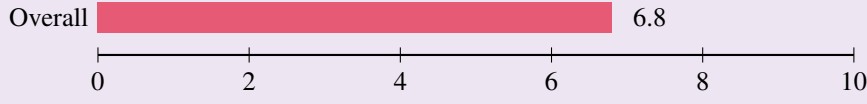

## G   Model and Training Parameters

In this section, we present the model configuration and training parameters for our framework in Tables 23 to 27.

| Parameter | Value |
| --- | --- |
| vocab_size | 128256 |
| max_position_embeddings | 131072 |
| hidden_size | 4096 |
| intermediate_size | 14336 |
| num_hidden_layers | 32 |
| num_attention_heads | 32 |
| num_key_value_heads | 8 |
| hidden_act | silu |
| initializer_range | 0.02 |
| rms_norm_eps | 1e-05 |
| pretraining_tp | 1 |
| use_cache | true |
| rope_theta | 500000.0 |
| rope_scaling.factor | 8.0 |
| rope_scaling.low_freq_factor | 1.0 |
| rope_scaling.high_freq_factor | 4.0 |
| rope_scaling.original_max_position_embeddings | 8192 |
| rope_scaling.rope_type | llama3 |
| head_dim | 128 |
| torch_dtype | bfloat16 |
| bos_token_id | 128000 |
| eos_token_id | [128001, 128008, 128009] |
| model_type | llama |
| architectures | LlamaForCausalLM |

Table 23: Llama 3.1 8B Instruct Configuration

| Parameter | Value |
| --- | --- |
| compute_environment | LOCAL_MACHINE |
| debug | false |
| distributed_type | DEEPSPEED |
| downcast_bf16 | no |
| enable_cpu_affinity | false |
| machine_rank | 0 |
| main_training_function | main |
| mixed_precision | bf16 |
| num_machines | 1 |
| num_processes | 2 |
| rdzv_backend | static |
| same_network | true |
| tpu_use_cluster | false |
| tpu_use_sudo | false |
| use_cpu | false |
| **deepspeed_config** | |
| gradient_accumulation_steps | 4 |
| gradient_clipping | 1.0 |
| offload_optimizer_device | cpu |
| offload_param_device | cpu |
| zero3_init_flag | false |
| zero3_save_16bit_model | true |
| zero_stage | 2 |

Table 24: Accelerate DeepSpeed Configuration

| Parameter | Value |
|---|---|
| compute_environment | LOCAL_MACHINE |
| debug | false |
| distributed_type | DEEPSPEED |
| downcast_bf16 | no |
| machine_rank | 0 |
| mixed_precision | bf16 |
| num_machines | 1 |
| num_processes | 2 |
| use_cpu | false |
| **deepspeed_config** | |
| gradient_accumulation_steps | 4 |
| gradient_clipping | 1.0 |
| offload_optimizer_device | none |
| offload_param_device | none |
| zero3_init_flag | false |
| zero3_save_16bit_model | true |
| zero_stage | 0 |

Table 25: Accelerate DeepSpeed Configuration for Knowledge Distillation

| Parameter | Value |
|---|---|
| **Model Configuration** | |
| model_name | Llama-3.1-8B-Instruct |
| Comments | Customized to do RL Reweighting for `Refit` and `Swift` |
| **Training Parameters** | |
| learning_rate | 3e-5 |
| num_train_epochs | 4 |
| per_device_train_batch_size | 8 |
| gradient_accumulation_steps | 4 |
| gradient_checkpointing | True |
| mixed_precision | bf16 |
| do_train | True |
| do_eval | False |
| logging_steps | 5 |
| logging_first_step | True |
| save_strategy | epoch |
| save_total_limit | 4 |
| **RL Configuration** | |
| dataset | From the listed datasets in this paper.json |
| rl_reweight | std |
| rl_reward_name | reward |
| use_custom_trainer | True |
| **Hardware Configuration** | |
| num_processes | 2 |
| num_machines | 1 |

Table 26: TRL Supervised Fine-Tuning Configuration with Customized model RL Reweighting for `Refit` and `Swift`

| Parameter | Value |
|---|---|
| **Model Configuration** | |
| teacher_model_path | `STaR-GATE` _last-checkpoint |
| student_model_name | meta-llama/Llama-3.1-8B-Instruct |
| student_layers | 8 |
| apply_lora_to_teacher | True |
| **LoRA Configuration** | |
| r | 8 |
| alpha | 16 |
| dropout | 0.05 |
| target_modules | q_proj, v_proj, k_proj, o_proj, gate_proj, up_proj, down_proj |
| **Distillation Parameters** | |
| distillation_alpha | 0.5 |
| distillation_temperature | 2.0 |
| **Training Parameters** | |
| learning_rate | 3e-6 |
| num_train_epochs | 2 |
| per_device_train_batch_size | 4 |
| gradient_accumulation_steps | 4 |
| gradient_checkpointing | True |
| mixed_precision | bf16 |
| do_train | True |
| do_eval | False |
| logging_steps | 5 |
| logging_first_step | True |
| save_strategy | epoch |
| save_total_limit | 4 |
| **Dataset Configuration** | |
| dataset | From the listed datasets in this paper |
| rl_reweight | SFT |
| use_custom_trainer | False |
| **Hardware Configuration** | |
| num_processes | 2 |
| num_machines | 1 |

Table 27: Knowledge Distillation Configuration with LoRA

