# OpenReview forum: "Offline RL by Reward-Weighted Fine-Tuning for Conversation Optimization"
_NeurIPS.cc/2025/Conference — NeurIPS 2025 poster_

### Official Review · Reviewer_rBkf · 2025-07-01

**Clarity:** 3
**Significance:** 3
**Originality:** 3
**Rating:** 5
**Confidence:** 4

**Summary:**

This paper introduces an offline RL algorithm to improve performance on QA tasks. Authors pose the QA problem as an RL problem, and proceed to derive a lower bound on policy gradient objective, which they also show is equivalent to a weighted SFT objective (i.e. increase likelihood of actions proportional to their rewards). They proceed to further improve it by standardizing the reward function and theoretically showing the algorithm still holds. Their methods outperformed various SOTA baselines on QA tasks.

**Questions:**

Thank you for your work! I enjoyed reading your paper.

My questions/suggestions are below:

* In your formulation, the value function doesn’t explicitly use the discount factor. You mentioned it being useful in the case when an agent tries to adaptively choose a number of turns. However, what about the general case? Did you use discounting in your experiments?
* The tables 1-9 use custom LLM-based metrics (i.e. reasoning, comprehensive etc.). How reliable are these metrics? I’d love to see a correlation with human raters, or possibly references to other research where they were used.
* I am not sure I understand how token level rewards are different from rewards used in your paper? My understanding is that modern RLHF approaches learn reward functions that measure the quality of a sequence of tokens (aka sentence). However, it seems you chose not to compare against methods that use token level rewards (PPO, GRPO). It’s a rather minor nit, if GRPO and PPO are not SOTA in QA domain.

**Ethical Concerns:**

["NO or VERY MINOR ethics concerns only"]

**Final Justification:**

Overall strong paper. I described its strengths and weaknesses in my official review. After rebuttal, the following still need to be addressed:

* Abstract and the main paper should explain that the method may not work for longer conversations (n>3).
* The Claude ablation is useful, but it's only done on the Arc dataset, and don't think adequately resolve the question of rating validity.

I am not convinced to raise my score. But given the overall quality and originality, this is an Accept paper.

**Limitations:**

Yes

**Quality:**

4

**Strengths And Weaknesses:**

# Quality

* (+) Well written paper that has references and clear comparisons to related works.
* (+) Explains their method from the first principles and builds intuition for the choice of offline RL.

# Clarity

* (+) Authors did a great job of using clear notation, progressively building-up concepts and clearly explaining math in their proofs.
* (-) Authors mentioned that rewards in their formulation assess the quality of the conversation. This is an ambiguous description, and the paper should include which rewards (i.e. which models/classifiers?) were used in the experiments.

# Significance

* (+) Authors pledged to release their customly generated QA dataset, which is helpful in advancing the field.
* (+) The offline RL proved to be helpful in other LLM tasks, and adapting it to QA can help make strong advances in the field.
* (-) The evaluation metrics only include automated metrics (accuracy and LLM-based). While accuracy is a reliable signal, LLM-based metrics are not validated (i.e. no prior use in literature, and no performance evaluation). Ideally, metrics from human raters are needed.

# Originality

* (+) The paper cleverly uses insights from other related works, and improves upon them. For example, using full reward signals for all of the trajectories (instead of just the best performing ones).

---

> ### Author Rebuttal · Authors · 2025-07-30
>
> We thank the reviewer for recognizing the originality, clarity, and quality of our work. Our rebuttal is below. We focus on major issues and will incorporate all comments of the reviewer in the next version of our paper. If you have additional concerns, please reach out to us to discuss them.
>
> ### **Q1: Rewards in Experiments**
>
> We do not discount rewards because all trajectories in Section 4 have length 3. We optimize the overall reward of the final answer by a GPT-4o judge at a fixed budget of 3 conversation turns. An ablation study with longer trajectories is in Appendix E. We show that when the number of training trajectories is fixed, optimization over a longer horizon is a harder problem.
>
> The LLM-as-a-judge was popularized by
>
> > Judging LLM-as-a-Judge with MT-Bench and Chatbot Arena (NeurIPS 2023)
>
> The system prompt, which we use in the GPT-4o judge to evaluate 5 aspects of language quality and obtain the overall reward, is borrowed from
>
> > Training LLM-based Tutors to Improve Student Learning Outcomes in Dialogues (AIED 2025)
>
> ### **Q2: LLM Evaluation Metrics**
>
> Similarly to closest related works [2, 13, 26], we do not conduct a human evaluation. To alleviate the concern that our evaluation is biased due to using a single GPT-4o judge, we follow the suggestion of Reviewer PkD3 and report results with a **Claude 4 Opus** judge on ARC dataset. The prompt is the same as in the GPT-4o judge.
>
> | Model | Accuracy | Thinking (%) | R Overall | R Accuracy | R Reasoning | R Comprehensive | R Pedagogic | R Confidence |
> | - | - | - | - | - | - | - | - | - |
> | SWiFt (ours) | 0.8667 ± 0.0310 | 97.5% ± 0.0% | 6.28 ± 0.19 | 8.76 ± 0.26 | 6.73 ± 0.18 | 5.79 ± 0.14 | 6.33 ± 0.19 | 7.24 ± 0.23 |
> | ReFit (ours) | 0.8583 ± 0.0318 | 98.3% ± 0.0% | 6.31 ± 0.20 | 8.69 ± 0.27 | 6.73 ± 0.19 | 5.78 ± 0.15 | 6.34 ± 0.19 | 7.25 ± 0.23 |
> | DPO | 0.7167 ± 0.0411 | 7.5% ± 0.0% | 4.29 ± 0.24 | 7.28 ± 0.37 | 4.50 ± 0.24 | 3.27 ± 0.18 | 3.30 ± 0.22 | 4.92 ± 0.28 |
> | STaR-GATE | 0.6990 ± 0.0270 | 90.0% ± 0.0% | 6.17 ± 0.17 | 7.48 ± 0.20 | 5.94 ± 0.16 | 5.22 ± 0.14 | 5.50 ± 0.16 | 7.11 ± 0.21 |
> | Base | 0.3772 ± 0.0146 | 75.1% ± 0.0% | 5.47 ± 0.12 | 7.32 ± 0.14 | 5.56 ± 0.11 | 5.80 ± 0.09 | 5.40 ± 0.11 | 5.92 ± 0.16 |
> | STaR-GATE-D | 0.8417 ± 0.0333 | 28.3% ± 0.0% | 4.22 ± 0.23 | 8.30 ± 0.33 | 4.17 ± 0.24 | 2.98 ± 0.18 | 3.06 ± 0.22 | 5.36 ± 0.28 |
> | StepDPO | 0.7167 ± 0.0411 | 6.7% ± 0.0% | 4.35 ± 0.24 | 7.22 ± 0.37 | 4.57 ± 0.24 | 3.31 ± 0.18 | 3.30 ± 0.23 | 5.08 ± 0.29 |
>
> The three best performing methods are the same as in Table 1: SWiFt, ReFit, and STaR-GATE-D. As in Table 1, SWiFt and ReFit attain much higher language quality LLM-as-a-judge scores than all baselines.
>
> ### **Q3: Token-Level Rewards**
>
> Our approach does not need to learn a reward model. This is because our objective is an expectation of the trajectory reward (for all tokens in it) multiplied by its log-probability, as shown in (4), and its gradient is an expectation of the trajectory reward multiplied by the gradient of its log-probability, as shown in (6). We discuss differences from PPO and GRPO at the end of Section 3.1. The closest related works [2, 13, 26] do not use them as baselines because they are not the state of the art in our domain, as you hinted on.

---

> > ### Comment · Reviewer_rBkf · 2025-08-08
> > **Response to rebuttal #1**
> >
> > Thank you for your response to my comments. I apologize for my late reply.
> >
> > Having n=3 limits (and no discounting for that case) in your setup makes sense, but if the method may not work for longer conversations, it should be stated in the abstract or the limitations. The Claude ablation is useful, but it's only done on the Arc dataset, and don't think adequately resolve the question of rating validity. I still think the paper is strong, but the rebuttal didn't convince me to raise my score.

---

> > > ### Author Response · Authors · 2025-08-09
> > >
> > > Dear Reviewer rBkf,
> > >
> > > Thank you for responding and maintaining your acceptance score. We agree with your suggestions and will modify the paper as follows:
> > >
> > > 1. We will clearly state in Abstract and limitations in Conclusions that we experiment with mostly shorter fixed-horizon conversations. Our goal is to maximize the reward of their final answers. Another possible setting are variable-length conversations. While our framework could be applied to these problems, through discounted rewards, we do not experiment with them. We will make this clear and suggest it as future work in Conclusions.
> > > 2. We agree that the Claude judge is not a replacement for a human study. Nevertheless, it shows the robustness of our policies to being judged by another LLM, because the policies are trained on GPT-4o rewards. We will apply the Claude judge to all problems in the paper (Tables 1-12) and include the corresponding tables in Appendix.
> > >
> > > Sincerely,
> > >
> > > Authors

---

### Official Review · Reviewer_7ZT6 · 2025-07-02

**Clarity:** 3
**Significance:** 3
**Originality:** 3
**Rating:** 5
**Confidence:** 2

**Summary:**

The paper proposes a reinforcement learning (RL) method to train models to ask clarifying questions in a QA task effectively. They propose an offline RL algorithm that optimizes a supervised fine-tuning objective weighted by the reward. They also propose a standardized reward to reduce variance and result in better policies. They conduct experiments on multiple QA benchmark datasets and show improved performance over related RL baselines. They also release a dataset of simulated multi-turn conversations for each of the benchmark dataset.

**Questions:**

- Will the code be released publicly to ensure correct reproducibility of the proposed algorithm?
- Do you do ablations on dataset size? For example, is performance robust to data size? what's the trade-off between computational cost and performance for different dataset size?
- Do you do human evaluation of the fine-tuned models' generations?
- What's the average number of turns in the generated multi-turn conversations of fine-tuned models? Do they only generate 3-turn conversations like the dataset they are trained with?
- Can you give illustrated examples of the generated clarifying questions so that we can see what the models actually learn?

**Ethical Concerns:**

["NO or VERY MINOR ethics concerns only"]

**Limitations:**

Yes

**Quality:**

3

**Strengths And Weaknesses:**

Strengths:
- the proposed offline RL algorithm is simple and novel; avoids complex token-level reward models or propensity scoring, and can be implemented as reward-weighted supervised fine-tuning, which can be applied to train LLMs.
- the idea of training models to ask clarifying questions in a QA task to improve its answer is interesting and impactful
- The paper is well-structured, making it easy to read. The algorithms are explained in a step-by-step manner that makes them easy to follow.
- the paper clearly defines the framing of the task as RL with mathematical derivations to justify the approach.
- the experiments are comprehensive with detailed and clear automatic evaluation metrics.

Weaknesses:
- although the proposed method is novel, it can be seen as a variation to some of the baselines it is compared to (although the baselines, as the authors mentioned, do not fully utilize the reward signals).
- the paper mentioned computational cost as potential limitation. However, some ablations on this aspect is missing, for example, experiments using different training data sizes.
- some of the mathematical derivations and lemmas can be explained more intuitively to improve the paper's readability
- the paper also mentioned that performance might depend on the quality of the training data, which is under explored in the paper -- only one way of sampling the trajectory (i.e., based on temperature) is explored.
- more critically, no human evaluation is conducted. It is unclear how the model will perform in real-world situation when interactions are with humans. No qualitative analysis or illustrated examples of the generated clarifying questions are given. The appendix only shows the (final) generated answer of the fine-tuned models, with the clarifying questions being the generic prompts from their simulated dataset. It will be good to have these examples to show what kinds of clarifying questions the models actually learn to generate. In addition, the average length of the generated multi-turn conversations of the fine-tuned models are not given. Do the models only generate 3-turn conversations like the dataset they are trained with?

---

> ### Author Rebuttal · Authors · 2025-07-30
>
> We thank the reviewer for recognizing the novelty of our work, clarity of writing, and comprehensive empirical evaluation. Our rebuttal is below. We focus on major issues and will incorporate all comments of the reviewer in the next version of our paper. If you have additional concerns, please reach out to us to discuss them.
>
> ### **Q1: Code Release**
>
> We are in the process of releasing dataset and code. We also wanted to stress that ReFit and SWiFt are extremely easy to implement because they reduce to reward-weighted fine-tuning.
>
> ### **Q2: Dataset Size Ablation**
>
> The computational costs of ReFit and SWiFt are linear in the dataset size, because both make a single pass over the logged dataset. We expect the reward to increase as the number of training trajectories increases. To show this, we conduct a dataset size ablation study on ARC dataset, in both thinking and standard modes. The results for ReFit and two different sample sizes (number of training trajectories) are reported below.
>
> **Thinking mode on ARC dataset**
>
> | Sample size | Accuracy | Thinking (%) | R Overall | R Accuracy | R Reasoning | R Comprehensive | R Pedagogic | R Confidence |
> | - | - | - | - | - | - | - | - | - |
> | 1000 | 0.7301 ± 0.0261 | 93.1% ± 0.0% | 6.75 ± 0.15 | 7.55 ± 0.19 | 7.03 ± 0.14 | 6.30 ± 0.12 | 6.51 ± 0.14 | 7.19 ± 0.19 |
> | 2000 | 0.7543 ± 0.0253 | 94.8% ± 0.0% | 6.88 ± 0.15 | 7.71 ± 0.18 | 7.18 ± 0.13 | 6.46 ± 0.10 | 6.67 ± 0.14 | 7.38 ± 0.18 |
>
> **Standard mode on ARC dataset**
>
> | Sample size | Accuracy | Thinking (%) | R Overall | R Accuracy | R Reasoning | R Comprehensive | R Pedagogic | R Confidence |
> | - | - | - | - | - | - | - | - | - |
> | 1000 | 0.6715 ± 0.0326 | 0.0% ± 0.0% | 6.64 ± 0.20 | 7.31 ± 0.24 | 6.97 ± 0.18 | 6.30 ± 0.15 | 6.54 ± 0.19 | 6.93 ± 0.26 |
> | 2000 | 0.7633 ± 0.0295 | 0.0% ± 0.0% | 7.25 ± 0.18 | 7.97 ± 0.22 | 7.50 ± 0.16 | 6.73 ± 0.14 | 7.14 ± 0.17 | 7.73 ± 0.23 |
>
> We observe that the accuracy and language quality metrics improve with more training data.
>
> ### **Q3: Human Evaluation**
>
> Similarly to closest related works [2, 13, 26], we do not conduct a human evaluation. To alleviate the concern that our evaluation is biased due to using a single GPT-4o judge, we follow the suggestion of Reviewer PkD3 and report results with a **Claude 4 Opus** judge on ARC dataset. The prompt is the same as in the GPT-4o judge.
>
> | Model | Accuracy | Thinking (%) | R Overall | R Accuracy | R Reasoning | R Comprehensive | R Pedagogic | R Confidence |
> | - | - | - | - | - | - | - | - | - |
> | SWiFt (ours) | 0.8667 ± 0.0310 | 97.5% ± 0.0% | 6.28 ± 0.19 | 8.76 ± 0.26 | 6.73 ± 0.18 | 5.79 ± 0.14 | 6.33 ± 0.19 | 7.24 ± 0.23 |
> | ReFit (ours) | 0.8583 ± 0.0318 | 98.3% ± 0.0% | 6.31 ± 0.20 | 8.69 ± 0.27 | 6.73 ± 0.19 | 5.78 ± 0.15 | 6.34 ± 0.19 | 7.25 ± 0.23 |
> | DPO | 0.7167 ± 0.0411 | 7.5% ± 0.0% | 4.29 ± 0.24 | 7.28 ± 0.37 | 4.50 ± 0.24 | 3.27 ± 0.18 | 3.30 ± 0.22 | 4.92 ± 0.28 |
> | STaR-GATE | 0.6990 ± 0.0270 | 90.0% ± 0.0% | 6.17 ± 0.17 | 7.48 ± 0.20 | 5.94 ± 0.16 | 5.22 ± 0.14 | 5.50 ± 0.16 | 7.11 ± 0.21 |
> | Base | 0.3772 ± 0.0146 | 75.1% ± 0.0% | 5.47 ± 0.12 | 7.32 ± 0.14 | 5.56 ± 0.11 | 5.80 ± 0.09 | 5.40 ± 0.11 | 5.92 ± 0.16 |
> | STaR-GATE-D | 0.8417 ± 0.0333 | 28.3% ± 0.0% | 4.22 ± 0.23 | 8.30 ± 0.33 | 4.17 ± 0.24 | 2.98 ± 0.18 | 3.06 ± 0.22 | 5.36 ± 0.28 |
> | StepDPO | 0.7167 ± 0.0411 | 6.7% ± 0.0% | 4.35 ± 0.24 | 7.22 ± 0.37 | 4.57 ± 0.24 | 3.31 ± 0.18 | 3.30 ± 0.23 | 5.08 ± 0.29 |
>
> The three best performing methods are the same as in Table 1: SWiFt, ReFit, and STaR-GATE-D. As in Table 1, SWiFt and ReFit attain much higher language quality LLM-as-a-judge scores than all baselines.
>
> ### **Q4: Trajectory Lengths**
>
> All trajectories in Section 4 have length 3. We optimize the overall reward of the final answer by a GPT-4o judge at a fixed budget of 3 conversation turns. An ablation study with longer trajectories is in Appendix E. We show that when the number of training trajectories is fixed, optimization over a longer horizon is a harder problem.
>
> ### **Q5: Changes in Learned Trajectories**
>
> The paper already shows changes in learned trajectories in two ways:
>
> * In Tables 1-12, we report 6 LLM-as-a-judge metrics: accuracy, reasoning ability, comprehensiveness, pedagogical value, confidence calibration, and overall score. In most cases, ReFit and SWiFt improve over Base (pre-trained LLM) in all metrics. This shows that the quality of the responses improves in multiple aspects.
> * In Figure 6 in Appendix H, we visualize the embeddings of Base and SWiFt responses in ARC and SciQA datasets by UMAP. We observe that the SWiFt responses are more diverse, because they correspond to a larger cluster after the UMAP projection. The clusters are centered at similar points.
>
> Examples of clarifying questions are below. For instance, the second turn response of Base to a multi-choice question
>
> *Several types of mechanisms in the body regulate the rate of chemical reactions in the digestive system. Among the most important are feedback systems, which control the release rate of: A. lipids. B. enzymes. C. nucleic acids. D. carbohydrates*
>
> in ARC dataset is
>
> *What do you think about how enzymes are involved in breaking down food in the digestive system? Can you think of a scenario where enzymes might be released or not released based on the presence of certain substances? This might help clarify the role of feedback systems in regulating enzyme release. Feedback systems in the body often work by monitoring the concentration of a particular substance in the bloodstream or a specific area. When the concentration of that substance reaches a certain threshold, the feedback system triggers a response to either increase or decrease the production or release of a substance. In the context of digestion, one possible scenario is that the presence of a certain type of food or nutrient triggers the release of enzymes to break it down.*
>
> This response is more verbose and less focused than that of SWiFt,
>
> *The answer is B.&lt;thinking&gt;To answer this question, I first considered each option and thought about what is known about each choice What do you know about these biological concepts? Hmm, I know enzymes play a pretty big role in the digestive process because they break things down—like proteins into amino acids or carbs into simpler sugars. They're kind of like the "workers" in the digestive system, right? But I'm a little unsure about the other options—lipids, nucleic acids, and carbohydrates. Are feedback systems connected to those, too? Like, do they regulate lipids or carbs directly, or is it more about how enzymes are released to handle those things?&lt;/thinking&gt;*

---

> > ### Comment · Reviewer_7ZT6 · 2025-08-05
> >
> > Thank you for addressing my questions. I will keep my score as is.

---

> > > ### Author Response · Authors · 2025-08-05
> > >
> > > Thank you for responding and maintaining the positive evaluation of the paper.

---

### Official Review · Reviewer_PkD3 · 2025-07-03

**Clarity:** 2
**Significance:** 2
**Originality:** 3
**Rating:** 5
**Confidence:** 3

**Summary:**

The paper recasts multi-turn clarifying question generation as an offline RL problem. The authors develop two objectives:

- ReFit: weight supervised fine-tuning loss by trajectory-level rewards.
- SWiFt: apply reward normalization before weighting to reduce variance.

Given that the expected reward gradient equals the expectation of reward × log p over demonstrations, training becomes weighted cross-entropy optimization. The contribution includes (i) a Jensen-inequality proof that the objective lower-bounds the standard RL value, and (ii) experiments on six benchmarks showing that ReFit/SWiFt outperform standard SFT and achieve competitive results with PPO-style methods at lower computational cost.

**Questions:**

1. **Baseline Completeness**
   Current baselines either (i) avoid asking clarification questions (SFT, DPO) or (ii) ignore reward magnitude (StepDPO, STaR-GATE).

   - Please add an SFT prompting baseline using identical three-question prompts with standard cross-entropy loss. This would isolate the contribution of reward weighting from the effect of enabling multi-turn conversation.

2. **"Thinking" Mode Justification**
   `<thinking>` tags are inserted although Llama-3-8B-Instruct is not chain-of-thought–trained, and Table 3 shows mixed or negative performance.
   - Provide an ablation (at equal token budget) demonstrating whether thinking mode actually improves accuracy. If it does not, please justify its inclusion.

3. **Reliance on GPT-4o Rewards**
   All rewards and automatic metrics are drawn from GPT4o, raising bias and domain-specificity concerns.

   - Report a small controlled study with an alternative evaluator (e.g., Claude-3.7 Sonnet or an open model has similar performance) to assess how sensitive the results are to the choice of reward model.
Update: This was provided in the author response.

**Ethical Concerns:**

["NO or VERY MINOR ethics concerns only"]

**Final Justification:**

The authors clarified several points in their response and also provided requested experimental results.

In light of this, several of the previously mentioned weaknesses no longer apply, and I am raising my score.

**Limitations:**

Yes, though somewhat minimal.

**Quality:**

3

**Strengths And Weaknesses:**

## Strengths

- **Algorithmic simplicity**: ReFit/SWiFt integrate seamlessly into existing SFT frameworks with minimal code changes, enhancing reproducibility.
- **Focused empirical contribution**: The work addresses a clear research question: whether trajectory-level reward weighting benefits clarification tasks with convincing experimental results.
- **Comprehensive analyses**: Normalization comparisons (ReFit -> SWiFt) and the reward-scale analysis effectively isolate the sources of performance gains.
- **Computational efficiency**: Training cost comparable to SFT without additional computational overhead.

## Weaknesses

- **Novelty**
  - The paper acknowledges that "our objective coincides with reward-weighted supervised fine-tuning" (l. 112–113) yet positions it as a novel "offline RL" algorithm.
- **Insufficient related work**
  - Section 5 provides limited comparison to relevant approaches. Notable work include "Reasoning with Language Model is Planning with World Model" and other planning-based methods that could provide meaningful performance comparisons.

---

> ### Author Rebuttal · Authors · 2025-07-30
>
> We thank the reviewer for a comprehensive review, which discusses both strengths and weaknesses of our work. This review contains several technical errors and we respectfully point them out in our rebuttal. We focus on major issues and will incorporate all comments of the reviewer in the next version of our paper. If you have additional concerns, please reach out to us to discuss them.
>
> ### **W1: Marginal Algorithmic Novelty**
>
> The claim that our objective reduces to that of REINFORCE is false. The REINFORCE objective is in (2). In plain English, it is the expected reward of trajectories generated by the optimized policy. The ReFit objective is in (4). It is the expected reward of trajectories multiplied by their log-probabilities. The trajectories are drawn from a previously logged dataset. The key properties of our objective are that it is a lower bound on (2) and that it can be optimized easily, because it reduces to reward-weighted fine-tuning. The main algorithmic novelty of our work is in this reduction.
>
> ### **W2: Related Works**
>
> Thank you for pointing out new related works. A comprehensive review of our related works is in Appendix B, where we mostly focus on learning to clarify and RL. We will expanded it with planning in LLMs at large. Specifically,
>
> > Language Models as Zero-Shot Planners: Extracting Actionable Knowledge for Embodied Agents (ICML 2022)
>
> is a seminal work on this topic,
>
> > Reasoning with Language Model is Planning with World Model (EMNLP 2023)
>
> searches for policies using Monte Carlo tree search, and
>
> > Describe, Explain, Plan and Select: Interactive Planning with LLMs Enables Open-World Multi-Task Agents (NeurIPS 2023)
>
> plans interatively and revises the plan based on reached sub-goals.
>
> ### **W3: Theoretical Foundation**
>
> The claim that (6) is derived using Jensen's inequality is false. (6) is the gradient of the ReFit objective, which is presented in (4). The derivation of (4) is in Lemma 1 and is also not a direct application of Jensen's inequality. We compate (4) to related works, including PPO and GRPO, in the last two paragraphs of Section 3.1.
>
> ### **W4: Insufficient Statistical Rigor**
>
> This evaluation contains several errors:
>
> * We never claim that all results are based on a single run. There is no footnote 6.
> * We do not experiment with QReCC dataset and never report BLEU.
>
> ### **Q1: Additional Baseline**
>
> We would appreciate if the reviewer clarified what SFT-Ask is. Can you please give us a link to the corresponding paper? We tried to find the algorithm online but we could not.
>
> One way of interpreting SFT-Ask is as doing SFT on all logged data, where each step in each trajectory is a separate data point. We wanted to point out that STaR-GATE, which we compare to in all experiments, can be viewed as SFT where each step in the most rewarding trajectories is a separate data point. This is a stronger and more established baseline.
>
> ### **Q2: Thinking Mode**
>
> Although Llama-3.1-8B-Instruct is not a reasoning model, we prompt it to reason within &lt;thinking&gt;&lt;/thinking&gt; tags. See Appendix D for examples of user messages that invoke thinking. We do not observe consistent improvements due to thinking because all models are compared at an equal token budget. Specifically, max_new_tokens = 150 tokens per response in all experiments. When the number of tokens is limited, the thinking does not always complete before max_new_tokens is reached.
>
> ### **Q3: GPT-4o Rewards**
>
> To alleviate the concern that our evaluation is biased due to using a single GPT-4o judge, we follow the suggestion of Reviewer PkD3 and report results with a **Claude 4 Opus** judge on ARC dataset. The prompt is the same as in the GPT-4o judge.
>
> | Model | Accuracy | Thinking (%) | R Overall | R Accuracy | R Reasoning | R Comprehensive | R Pedagogic | R Confidence |
> | - | - | - | - | - | - | - | - | - |
> | SWiFt (ours) | 0.8667 ± 0.0310 | 97.5% ± 0.0% | 6.28 ± 0.19 | 8.76 ± 0.26 | 6.73 ± 0.18 | 5.79 ± 0.14 | 6.33 ± 0.19 | 7.24 ± 0.23 |
> | ReFit (ours) | 0.8583 ± 0.0318 | 98.3% ± 0.0% | 6.31 ± 0.20 | 8.69 ± 0.27 | 6.73 ± 0.19 | 5.78 ± 0.15 | 6.34 ± 0.19 | 7.25 ± 0.23 |
> | DPO | 0.7167 ± 0.0411 | 7.5% ± 0.0% | 4.29 ± 0.24 | 7.28 ± 0.37 | 4.50 ± 0.24 | 3.27 ± 0.18 | 3.30 ± 0.22 | 4.92 ± 0.28 |
> | STaR-GATE | 0.6990 ± 0.0270 | 90.0% ± 0.0% | 6.17 ± 0.17 | 7.48 ± 0.20 | 5.94 ± 0.16 | 5.22 ± 0.14 | 5.50 ± 0.16 | 7.11 ± 0.21 |
> | Base | 0.3772 ± 0.0146 | 75.1% ± 0.0% | 5.47 ± 0.12 | 7.32 ± 0.14 | 5.56 ± 0.11 | 5.80 ± 0.09 | 5.40 ± 0.11 | 5.92 ± 0.16 |
> | STaR-GATE-D | 0.8417 ± 0.0333 | 28.3% ± 0.0% | 4.22 ± 0.23 | 8.30 ± 0.33 | 4.17 ± 0.24 | 2.98 ± 0.18 | 3.06 ± 0.22 | 5.36 ± 0.28 |
> | StepDPO | 0.7167 ± 0.0411 | 6.7% ± 0.0% | 4.35 ± 0.24 | 7.22 ± 0.37 | 4.57 ± 0.24 | 3.31 ± 0.18 | 3.30 ± 0.23 | 5.08 ± 0.29 |
>
> The three best performing methods are the same as in Table 1: SWiFt, ReFit, and STaR-GATE-D. As in Table 1, SWiFt and ReFit attain much higher language quality LLM-as-a-judge scores than all baselines.

---

> > ### Comment · Reviewer_PkD3 · 2025-08-09
> > **Reply to Author Response**
> >
> > Many thanks for your detailed response and for clarifying several points.
> >
> > W1 & W3: Thanks for clarifying these misunderstandings. I will revise the review accordingly.
> >
> > W2: Thank you for including the additional related work.
> >
> > W4: You are indeed correct. This was due to another paper referenced while reviewing this paper. I will revise the review accordingly.
> >
> > Q1: I was not referring to a specific published method, but rather to a simple SFT baseline. I now see that STaR-GATE already serves as a sufficient example of such a baseline.
> >
> > Q2: I understand the rationale for using thinking tags. My point was about potential confusion compared to standard short-CoT prompting. In my experience, thinking tags are more prevalent in models trained for long-form reasoning. Therefore, using them in a short-CoT context with a 150-token limit and a model not trained for long CoT feels unconventional.
> >
> > Q3: I appreciate the Claude 4 Opus judge experiments, which address my concerns about dependence on the reward model.
> >
> > These clarifications address my main concerns, and I am therefore raising my score.

---

### Official Review · Reviewer_GFfE · 2025-07-13

**Clarity:** 2
**Significance:** 2
**Originality:** 3
**Rating:** 4
**Confidence:** 4

**Summary:**

The paper "Learning to Clarify by Reinforcement Learning Through Reward-Weighted Fine-Tuning" presents a novel approach to enhance question-answering (QA) agents by enabling them to ask clarifying questions. The authors propose a reinforcement learning (RL) framework that simulates conversations with clarifying questions and learns from them. To make RL feasible, they introduce offline RL objectives that resemble reward-weighted supervised fine-tuning (SFT), which can be easily optimized in large language models. This approach contrasts with existing methods that rely on SFT and direct preference optimization, which involve additional hyper-parameters and do not directly optimize rewards. The paper demonstrates empirical improvements over strong baselines like STaR-GATE on datasets like ARC, MMLU, OpenBookQA etc.

**Questions:**

The results are decent in the paper; however it lacks analysis, there are multiple components in the paper which haven't been ablated. Similarly it's not clear why sometimes ReFit is better vs why sometimes  SWiFt is better?

**Ethical Concerns:**

["NO or VERY MINOR ethics concerns only"]

**Final Justification:**

I maintain my rating, the oveall work is good but there are issues with  Inconsistent Improvements and Insufficient Statistical Rigor as mentioned by other reviewers as well.

**Limitations:**

yes

**Quality:**

3

**Strengths And Weaknesses:**

Strengths:
1) The paper address a very novel problem of Question answering (QA) agents and proposes method to improve the asking of clarifying questions. Overall the method is novel and has good intutions.
2) The paper shows a lot of results across multiple datasets like  ARC, MMLU, OpenBookQA and shows improvements over most of them.

Weakness:
1) I would like to see the qualitative analysis of empirical variance which is measured in eqn6 . How much of it is reduce after Standardized Reward-Weighted Fine-Tuning.
2) Is there any analysis on trajectories or visual understanding of how they are behaving and changing with optimization process?
3)  The results are not really consistent and in datasets mathdial and OpenbookQA ; different methods are better. Also any explanation on why sometimes ReFit is better vs why sometimes  SWiFt is better?
4) The writing needs to be bit more clear, for ex. In algorithm1 how its the logged dataset generated?  Needs more details about dataset generation and sample examples.
5) I can see more examples in the appendix, but I would suggest to have one solid example in the main paper as well for better visualization.

---

> ### Author Rebuttal · Authors · 2025-07-30
>
> We thank the reviewer for recognizing the novelty of our work and its comprehensive empirical evaluation. Our rebuttal is below. We focus on major issues and will incorporate all comments of the reviewer in the next version of our paper. If you have additional concerns, please reach out to us to discuss them.
>
> ### **W1: Improvements in SWiFt Objective**
>
> Our analysis builds on Sections 1 and 2 of
>
> > A Short Note on Concentration Inequalities for Random Vectors with SubGaussian Norm (arXiv 2019)
>
> Take (6) and let
>
> $$g(x, \tau_n; \theta)
> = \sum_{t = 1}^n \nabla \log \pi(a_t \mid x, \tau_{t - 1}; \theta)$$
>
> be the random gradient inside the expectation, for random $x$ and $\tau_n$. Suppose that
>
> $$||g(x, \tau_n; \theta)||_2 \leq \sigma$$
>
> holds for any $x$, $\tau_n$, and $\theta$. Then $g(x, \tau_n; \theta)$ is a $\sigma$-norm-sub-Gaussian vector (Lemma 1 in the note). Let all rewards be non-negative (Section 2) and $r_\max = \max_{x, \tau_n} r(x, \tau_n)$ be the maximum reward. Then $r(x, \tau_n) g(x, \tau_n; \theta)$ is $(r_\max \sigma)$-norm-sub-Gaussian. The average of such random vectors concentrates at (6) in the $L_2$-norm at a rate proportional to their sub-Gaussianity parameter $r_\max \sigma$ (Definition 3 in the note).
>
> Let $\mu_x = \mathbb{E}[r(x, \tau_n) \mid x]$. Since the rewards are non-negative, $|r(x, \tau_n) - \mu_x| \leq r_\max$ holds for any $x$ and $\tau_n$. Therefore, $(r(x, \tau_n) - \mu_x) g(x, \tau_n; \theta)$ is at most $(r_\max \sigma)$-norm-sub-Gaussian, and the average of such random vectors concentrates faster than without subtracting the mean. Moreover, since
>
> $$\mathbb{E}[r(x, \tau_n) g(x, \tau_n; \theta)]
> = \mathbb{E}[(r(x, \tau_n) - \mu_x) g(x, \tau_n; \theta)]$$
>
> the optimized function has not changed but we get a **higher statistical efficiency** in estimating the gradient.
>
> To the best of our knowledge, the normalization by $\sigma_x^2 = \text{var}[r(x, \tau_n) \mid x]$ in
>
> $$\mathbb{E}\left[\frac{r(x, \tau_n) - \mu_x}{\sigma_x}
> g(x, \tau_n; \theta)\right]$$
>
> is hard to analyze because it changes the gradient. It tends to help in practice because it renormalizes the variances of rewards across all contexts $x$. Therefore, the learned policy improves more equally in all contexts without tuning the learning rate per context. This trick has been popularized by GRPO.
>
> ### **W2: Changes in Learned Trajectories**
>
> The paper already shows changes in learned trajectories in two ways:
>
> * In Tables 1-12, we report 6 LLM-as-a-judge metrics: accuracy, reasoning ability, comprehensiveness, pedagogical value, confidence calibration, and overall score. In most cases, ReFit and SWiFt improve over Base (pre-trained LLM) in all metrics. This shows that the quality of the responses improves in multiple aspects.
> * In Figure 6 in Appendix H, we visualize the embeddings of Base and SWiFt responses in ARC and SciQA datasets by UMAP. We observe that the SWiFt responses are more diverse, because they correspond to a larger cluster after the UMAP projection. The clusters are centered at similar points.
>
> The higher quality of SWiFt responses can also be observed on concrete examples. For instance, the final response of Base to a multi-choice question
>
> *Conifers are the most prevalent type of what plant group?*
>
> in SciQA dataset is
>
> *Gymnosperms, as shown in the diagram below, are the correct choice. They include conifers, cycads, and ginkgoes. They have seeds that are not enclosed in a fruit.&lt;/thinking&gt;*
>
> This response is worse, both in terms of format and content, than that of SWiFt,
>
> *The answer is B.&lt;thinking&gt;The question is asking about plant groups. The options provided include theropods, gymnosperms, algae, and angiosperms. Theropods are not plants, they are a subgroup of dinosaurs. Algae are primitive aquatic plants, but they are not the correct answer. Angiosperms are flowering plants, but the question is looking for the most prevalent type of plant group, not the most specific.&lt;/thinking&gt;*
>
> ### **W3: Inconsistent Improvements**
>
> We reviewed Tables 1-12. The only table where the accuracy of SWiFt is significantly worse than that of a baseline is Table 12. On the same dataset in thinking mode (Table 6), the accuracy of SWiFt is comparable to the best baseline in Table 12. In summary, SWiFt is the best or near-best method on all datasets.
>
> We also note that ReFit has a higher accuracy than SWiFt in Table 2 only. In this case, the accuracies of both methods are similar (within 0.01) and so are the language metrics. If we set 0.01 as a significance threshold, the accuracy of SWiFt would be always higher than that of ReFit when the accuracies are at least 0.01 apart.
>
> Finally, Reviewer PkD3 suggested that we experiment with another judge than GPT-4o. Our results with a **Claude 4 Opus** judge on ARC dataset are reported below. The prompt is the same as in the GPT-4o judge.
>
> | Model | Accuracy | Thinking (%) | R Overall | R Accuracy | R Reasoning | R Comprehensive | R Pedagogic | R Confidence |
> | - | - | - | - | - | - | - | - | - |
> | SWiFt (ours) | 0.8667 ± 0.0310 | 97.5% ± 0.0% | 6.28 ± 0.19 | 8.76 ± 0.26 | 6.73 ± 0.18 | 5.79 ± 0.14 | 6.33 ± 0.19 | 7.24 ± 0.23 |
> | ReFit (ours) | 0.8583 ± 0.0318 | 98.3% ± 0.0% | 6.31 ± 0.20 | 8.69 ± 0.27 | 6.73 ± 0.19 | 5.78 ± 0.15 | 6.34 ± 0.19 | 7.25 ± 0.23 |
> | DPO | 0.7167 ± 0.0411 | 7.5% ± 0.0% | 4.29 ± 0.24 | 7.28 ± 0.37 | 4.50 ± 0.24 | 3.27 ± 0.18 | 3.30 ± 0.22 | 4.92 ± 0.28 |
> | STaR-GATE | 0.6990 ± 0.0270 | 90.0% ± 0.0% | 6.17 ± 0.17 | 7.48 ± 0.20 | 5.94 ± 0.16 | 5.22 ± 0.14 | 5.50 ± 0.16 | 7.11 ± 0.21 |
> | Base | 0.3772 ± 0.0146 | 75.1% ± 0.0% | 5.47 ± 0.12 | 7.32 ± 0.14 | 5.56 ± 0.11 | 5.80 ± 0.09 | 5.40 ± 0.11 | 5.92 ± 0.16 |
> | STaR-GATE-D | 0.8417 ± 0.0333 | 28.3% ± 0.0% | 4.22 ± 0.23 | 8.30 ± 0.33 | 4.17 ± 0.24 | 2.98 ± 0.18 | 3.06 ± 0.22 | 5.36 ± 0.28 |
> | StepDPO | 0.7167 ± 0.0411 | 6.7% ± 0.0% | 4.35 ± 0.24 | 7.22 ± 0.37 | 4.57 ± 0.24 | 3.31 ± 0.18 | 3.30 ± 0.23 | 5.08 ± 0.29 |
>
> This experiment shows that SWiFt has a slightly higher accuracy than ReFit. Both methods attain much higher language quality LLM-as-a-judge scores than all baselines.
>
> ### **W4: Additional Clarifications**
>
> Data logging for ReFit is described in lines 169-170 and data logging for SWiFt is described in lines 212-218. To make this clear, we will include pseudo-code of both procedures in the next version of the paper.
>
> ### **W5: Conversation Example in Main Paper**
>
> In the next version of the paper, we will include one conversation example from Appendix D in the main paper.

---

### Comment · Area_Chair_YvFp · 2025-08-06

Dear reviewers,

This is a reminder that the end of author-reviewer discussion period is **Aug. 8**. Please do carefully read all other reviews and the author responses; and discuss openly with the authors, especially on your own questions that the authors addressed.

Best,
AC

---

### Note · Authors · 2025-08-12

Dear Reviewers and Area Chair,

This is a brief summary of our reviews and engagement with the reviewers. We believe that our paper was well received: accept with confidence 4 (**Reviewer rBkf**), accept with confidence 2 (**Reviewer 7ZT6**), borderline accept with confidence 4 (**Reviewer GFfE**), and borderline reject with confidence 3 (**Reviewer PkD3**). The reviewers engaged with us as follows:

* **Reviewer GFfE** acknowledged our rebuttal.
* **Reviewer PkD3** initially recommended borderline rejection of our paper and increased their score after the rebuttal.
* **Reviewer 7ZT6** responded to us and maintained their score after the rebuttal.
* **Reviewer rBkf** responded to us and maintained their score after the rebuttal. They had additional suggestions and we agreed to modify the paper accordingly.

Thank you for the reviews and engaging with us.

Sincerely,

Authors

---

### Decision · Program_Chairs · 2025-09-17

**Decision:**

Accept (poster)

**Comment:**

The paper proposes ReFit and SWiFt, two offline RL methods for training QA agents to ask clarifying questions. By reducing the RL objective to reward-weighted supervised fine-tuning, the methods are simple, efficient, and theoretically grounded. Experiments on benchmarks such as ARC, MMLU, and OpenBookQA show consistent improvements over SFT, DPO, and STaR-GATE.

**Strengths**

- Novel framing of clarifying question generation via offline RL.

- Methods are easy to implement, efficient, and reproducible.

- Strong empirical performance across multiple QA tasks.

- Clear presentation with promised code and dataset release.

**Weaknesses:**

- Improvements are not uniform across datasets.

- Heavy reliance on LLM-as-judge metrics; no human evaluation.

- Scope limited to fixed 3-turn conversations.

- Limited qualitative analysis of generated clarifying questions.

--

Reviewers raised concerns about inconsistent results, evaluation design, and missing qualitative analysis. The authors provided clarifications, dataset size ablations, and results with an alternative evaluator (Claude), which addressed some but not all concerns. Reviewers maintained overall positive or borderline-positive scores. Balancing strengths and weaknesses, I recommend acceptance.